# Deconditional Downscaling with Gaussian Processes

**Siu Lun Chau**[*†]
University of Oxford

**Shahine Bouabid**[*†]
University of Oxford

**Dino Sejdinovic**[†]
University of Oxford

## Abstract

Refining low-resolution (LR) spatial fields with high-resolution (HR) information, often known as *statistical downscaling*, is challenging as the diversity of spatial datasets often prevents direct matching of observations. Yet, when LR samples are modeled as aggregate conditional means of HR samples with respect to a mediating variable that is globally observed, the recovery of the underlying fine-grained field can be framed as taking an "inverse" of the conditional expectation, namely a *deconditioning problem*. In this work, we propose a Bayesian formulation of deconditioning which naturally recovers the initial reproducing kernel Hilbert space formulation from Hsu and Ramos [1]. We extend deconditioning to a downscaling setup and devise efficient conditional mean embedding estimator for multiresolution data. By treating conditional expectations as inter-domain features of the underlying field, a posterior for the latent field can be established as a solution to the deconditioning problem. Furthermore, we show that this solution can be viewed as a two-staged vector-valued kernel ridge regressor and show that it has a minimax optimal convergence rate under mild assumptions. Lastly, we demonstrate its proficiency in a synthetic and a real-world atmospheric field downscaling problem, showing substantial improvements over existing methods.

## 1 Introduction

Spatial observations often operate at limited resolution due to practical constraints. For example, remote sensing atmosphere products [2, 3, 4, 5] provide measurement of atmospheric properties such as cloud top temperatures and optical thickness, but only at a low resolution. Devising methods to refine low-resolution (LR) variables for local-scale analysis thus plays a crucial part in our understanding of the anthropogenic impact on climate

When high-resolution (HR) observations of different covariates are available, details can be instilled into the LR field for refinement. This task is referred to as *statistical downscaling* or *spatial disaggregation* and models LR observations as the aggregation of an unobserved underlying HR field. For example, multispectral optical satellite imagery [6, 7] typically comes at higher resolution than atmospheric products and can be used to refine the latter.

Statistical downscaling has been studied in various forms, notably giving it a probabilistic treatment [8, 9, 10, 11, 12, 13], in which Gaussian processes (GP) [14] are typically used in conjunction with a sparse variational formulation [15] to recover the underlying unobserved HR field. Our approach follows this line of research where we do not observe data from the underlying HR groundtruth field. On the other hand, deep neural network (DNN) based approaches [16, 17, 18] study this problem from a different setting, where they often assume that both HR and LR matched observations are available for training. Then, their approaches follow a standard supervised learning setting in learning a mapping between different resolutions.

---

[*]Indicates equal contribution

[†]Department of Statistics, Oxford, UK, OX1 3LB. <siu.chau@stats.ox.ac.uk, shahine.bouabid@stats.ox.ac.uk, dino.sejdinovic@stats.ox.ac.uk>

35th Conference on Neural Information Processing Systems (NeurIPS 2021).

However, both lines of existing methods require access to bags of HR covariates that are paired with aggregated targets, which in practice might be infeasible. For example, the multitude of satellites in orbit not only collect snapshots of the atmosphere at different resolutions, but also from different places and at different times, such that these observations are not jointly observed. To overcome this limitation, we propose to consider a more flexible *mediated statistical downscaling* setup that only requires indirect matching between LR and HR covariates through a mediating field. We assume that this additional field can be easily observed, and matched separately with both HR covariates and aggregate LR targets. We then use this third-party field to mediate learning and downscale our unmatched data. In our motivating application, climate simulations [19, 20, 21] based on physical science can serve as a mediating field since they provide a comprehensive spatiotemporal coverage of meteorological variables that can be matched to both LR and HR covariates.

Formally, let $^b\boldsymbol{x} = \{x^{(1)}, \ldots, x^{(n)}\} \subset \mathcal{X}$ be a general notation for bags of HR covariates, $f : \mathcal{X} \to \mathbb{R}$ the field of interest we wish to recover and $\tilde{z}$ the LR aggregate observations from the field $f$. We suppose that $^b\boldsymbol{x}$ and $\tilde{z}$ are unmatched, but that there exists mediating covariates $y, \tilde{y} \in \mathcal{Y}$, such that $(^b\boldsymbol{x}, y)$ are jointly observed and likewise for $(\tilde{y}, \tilde{z})$ as illustrated in Figure 1. We assume the following aggregation observation model $\tilde{z} = \mathbb{E}_X[f(X)|Y = \tilde{y}] + \varepsilon$ with some noise $\varepsilon$. Our goal in mediated statistical downscaling is then to estimate $f$ given $(^b\boldsymbol{x}, y)$ and $(\tilde{y}, \tilde{z})$, which corresponds to the *deconditioning* problem introduced in [1].

Motivated by applications in likelihood-free inference and task-transfer regression, Hsu and Ramos [1] first studied the deconditioning problem through the lens of reproducing kernel Hilbert space (RKHS) and introduced the framework of *Deconditional Mean Embeddings* (DME) as its solution.

In this work, we first propose a Bayesian formulation of deconditioning that results into a much simpler and elegant way to arrive to the DME-based estimator of Hsu and Ramos [1], using the conditional expectations of $f$. Motivated by our mediated statistical downscaling problem, we then extend deconditioning to a multi-resolution setup and bridge the gap between DMEs and existing probabilistic statistical downscaling methods [9]. By placing a GP prior on the sought field $f$, we obtain a posterior distribution of the downscaled field as a principled Bayesian solution to the downscaling task on indirectly matched data. For scalability, we provide a tractable variational inference approximation and an alternative approximation to the conditional mean operator (CMO) [22] to speed up computation for large multi-resolution datasets.

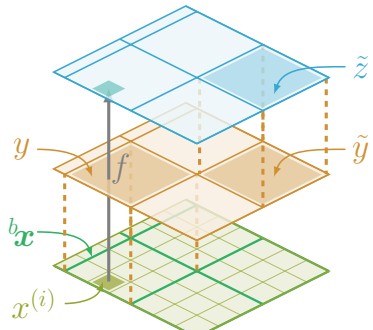

Figure 1: The LR response $\tilde{z}$ (blue) and the bag HR covariates $^b\boldsymbol{x}$ (green) are unmatched. The downscaling is mediated through bag-level LR covariates $y$ and $\tilde{y}$ (orange).

From a theoretical stand point, we further develop the framework of DMEs by establishing it as a two-staged vector-valued regressor with a natural reconstruction loss that mirrors Grünewälder et al. [23]'s work on conditional mean embeddings. This perspective allows us to leverage distribution regression theory from [24, 25] and obtain convergence rate results for the deconditional mean operator (DMO) estimator. Under mild assumptions, we obtain conditions under which this rate is a minimax optimal in terms of statistical-computational efficiency.

Our contributions are summarized as follows:

- We propose a Bayesian formulation of the deconditioning problem of Hsu and Ramos [1]. We establish its posterior mean as a DME-based estimate and its posterior covariance as a gauge of the deconditioning quality.

- We extend deconditioning to a multi-resolution setup in the context of the mediated statistical downscaling problem and bridge the gap with existing probabilistic statistical downscaling methods. Computationally efficient algorithms are devised.

- We demonstrate that the DMO estimate minimises a two-staged vector-valued regression and derive its convergence rate under mild assumptions, with conditions for minimax optimality.

- We benchmark our model against existing methods for statistical downscaling tasks in climate science, on both synthetic and real-world multi-resolution atmospheric fields data, and show improved performance.

## 2 Background Materials

### 2.1 Notations

Let $X, Y$ be a pair of random variables taking values in non-empty sets $\mathcal{X}$ and $\mathcal{Y}$, respectively. Let $k : \mathcal{X} \times \mathcal{X} \to \mathbb{R}$ and $\ell : \mathcal{Y} \times \mathcal{Y} \to \mathbb{R}$ be positive definite kernels. The closure of the span of their canonical feature maps $k_x := k(x, \cdot)$ and $\ell_y := \ell(y, \cdot)$ for $x \in \mathcal{X}$ and $y \in \mathcal{Y}$ respectively induces RKHS $\mathcal{H}_k \subseteq \mathbb{R}^{\mathcal{X}}$ and $\mathcal{H}_\ell \subseteq \mathbb{R}^{\mathcal{Y}}$ endowed with inner products $\langle \cdot, \cdot \rangle_k$ and $\langle \cdot, \cdot \rangle_\ell$.

We observe realizations $^b\mathcal{D}_1 = \{^b\boldsymbol{x}_j, y_j\}_{j=1}^N$ of bags $^b\boldsymbol{x}_j = \{x_j^{(i)}\}_{i=1}^{n_j}$ from conditional distribution $\mathbb{P}_{X|Y=y_j}$, with bag-level covariates $y_j$ sampled from $\mathbb{P}_Y$. We concatenate them into vectors $\mathbf{x} := \begin{bmatrix} ^b\boldsymbol{x}_1 & \dots & ^b\boldsymbol{x}_N \end{bmatrix}^\top$ and $\mathbf{y} := \begin{bmatrix} y_1 & \dots & y_N \end{bmatrix}^\top$.

For simplicity, our exposition will use the notation without bagging – i.e. where $\mathcal{D}_1 = \{x_j, y_j\}_{j=1}^N$ – when the generality of our contributions will be relevant to the theory of deconditioning from Hsu and Ramos [1], in Sections 2, 3 and 4. We will come back to a bagged dataset formalism in Sections 3.3 and 5, which corresponds to our motivating application of mediated statistical downscaling.

With an abuse of notation, we define feature matrices by stacking feature maps along columns as $\boldsymbol{\Phi}_{\mathbf{x}} := \begin{bmatrix} k_{x_1} & \dots & k_{x_N} \end{bmatrix}$ and $\boldsymbol{\Psi}_{\mathbf{y}} := \begin{bmatrix} \ell_{y_1} & \dots & \ell_{y_N} \end{bmatrix}$ and we denote Gram matrices as $\mathbf{K_{xx}} = \boldsymbol{\Phi}_{\mathbf{x}}^\top \boldsymbol{\Phi}_{\mathbf{x}} = [k(x_i, x_j)]_{1 \le i,j \le N}$ and $\mathbf{L_{yy}} = \boldsymbol{\Psi}_{\mathbf{y}}^\top \boldsymbol{\Psi}_{\mathbf{y}} = [\ell(y_i, y_j)]_{1 \le i,j \le N}$. The notation abuse $(\cdot)^\top(\cdot)$ is a shorthand for the elementwise RKHS inner products when it is clear from the context.

Let $Z$ denote the real-valued random variable stemming from the noisy conditional expectation of some unknown latent function $f : \mathcal{X} \to \mathbb{R}$, as $Z = \mathbb{E}_X[f(X)|Y] + \varepsilon$. We suppose one observes another set of realizations $\mathcal{D}_2 = \{\tilde{y}_j, \tilde{z}_j\}_{j=1}^M$ from $\mathbb{P}_{YZ}$, which is sampled independently from $\mathcal{D}_1$. Likewise, we stack observations into vectors $\tilde{\mathbf{y}} := \begin{bmatrix} \tilde{y}_1 & \dots & \tilde{y}_M \end{bmatrix}^\top$, $\tilde{\mathbf{z}} := \begin{bmatrix} \tilde{z}_1 & \dots & \tilde{z}_M \end{bmatrix}^\top$ and define feature map $\boldsymbol{\Psi}_{\tilde{\mathbf{y}}} := \begin{bmatrix} \ell_{\tilde{y}_1} & \dots & \ell_{\tilde{y}_M} \end{bmatrix}$.

### 2.2 Conditional and Deconditional Kernel Mean Embeddings

**Marginal and Joint Mean Embeddings**  Kernel mean embeddings of distributions provide a powerful framework for representing and manipulating distributions without specifying their parametric form [22, 26]. The marginal mean embedding of measure $\mathbb{P}_X$ is defined as $\mu_X := \mathbb{E}_X[k_X] \in \mathcal{H}_k$ and corresponds to the Riesz representer of expectation functional $f \mapsto \mathbb{E}_X[f(X)]$. It can hence be used to evaluate expectations $\mathbb{E}_X[f(X)] = \langle f, \mu_X \rangle_k$. If the mapping $\mathbb{P}_X \mapsto \mu_X$ is injective, the kernel $k$ is said to be characteristic [27], a property satisfied for the Gaussian and Matérn kernels on $\mathbb{R}^d$ [28]. In practice, Monte Carlo estimator $\hat{\mu}_X := \frac{1}{N}\sum_{i=1}^N k_{x_i}$ provides an unbiased estimate of $\mu_X$ [29].

Extending this rationale to embeddings of joint distributions, we define $C_{YY} := \mathbb{E}_Y[\ell_Y \otimes \ell_Y] \in \mathcal{H}_\ell \otimes \mathcal{H}_\ell$ and $C_{XY} := \mathbb{E}_{X,Y}[k_X \otimes \ell_Y] \in \mathcal{H}_k \otimes \mathcal{H}_\ell$, which can be identified with the cross-covariance operators between Hilbert spaces $C_{YY} : \mathcal{H}_\ell \to \mathcal{H}_\ell$ and $C_{XY} : \mathcal{H}_\ell \to \mathcal{H}_k$. They correspond to the Riesz representers of bilinear forms $(g, g') \mapsto \mathrm{Cov}(g(Y), g'(Y)) = \langle g, C_{YY}g' \rangle_\ell$ and $(f, g) \mapsto \mathrm{Cov}(f(X), g(Y)) = \langle f, C_{XY}g \rangle_k$. As above, empirical estimates are obtained as $\hat{C}_{YY} = \frac{1}{N}\boldsymbol{\Psi}_{\mathbf{y}}\boldsymbol{\Psi}_{\mathbf{y}}^\top = \frac{1}{N}\sum_{i=1}^N \ell_{y_i} \otimes \ell_{y_i}$ and $\hat{C}_{XY} = \frac{1}{N}\boldsymbol{\Phi}_{\mathbf{x}}\boldsymbol{\Psi}_{\mathbf{y}}^\top = \frac{1}{N}\sum_{i=1}^N k_{x_i} \otimes \ell_{y_i}$. Again, notation abuse $(\cdot)(\cdot)^\top$ is a shorthand for element-wise tensor products when clear from context.

**Conditional Mean Embeddings**  Similarly, one can introduce RKHS embeddings for conditional distributions referred to as *Conditional Mean Embeddings* (CME). The CME of conditional probability measure $\mathbb{P}_{X|Y=y}$ is defined as $\mu_{X|Y=y} := \mathbb{E}_X[k_X|Y = y] \in \mathcal{H}_k$. As introduced by Fukumizu et al. [27], it is common to formulate conditioning in terms of a Hilbert space operator $C_{X|Y} : \mathcal{H}_\ell \to \mathcal{H}_k$ called the *Conditional Mean Operator* (CMO). $C_{X|Y}$ satisfies by definition $C_{X|Y}\ell_y = \mu_{X|Y=y}$ and $C_{X|Y}^\top f = \mathbb{E}_X[f(X)|Y = \cdot], \forall f \in \mathcal{H}_k$, where $C_{X|Y}^\top$ de-

notes the adjoint of $C_{X|Y}$. Plus, the CMO admits expression $C_{X|Y} = C_{XY}C_{YY}^{-1}$, provided $\ell_y \in \text{Range}(C_{YY})$, $\forall y \in \mathcal{Y}$ [26, 27]. Song et al. [30] show that a nonparametric empirical form of the CMO writes

$$\hat{C}_{X|Y} = \mathbf{\Phi}_{\mathbf{x}}(\mathbf{L}_{\mathbf{yy}} + N\lambda\mathbf{I}_N)^{-1}\mathbf{\Psi}_{\mathbf{y}}^{\top}, \tag{1}$$

where $\lambda > 0$ is some regularisation ensuring the empirical operator is globally defined and bounded.

As observed by Grünewälder et al. [23], since $C_{X|Y}$ defines a mapping from $\mathcal{H}_\ell$ to $\mathcal{H}_k$, it can be interpreted as the solution to a vector-valued regression problem. This perspective enables derivation of probabilistic convergence bounds on the empirical CMO estimator [23, 25].

**Deconditional Mean Embeddings**    Introduced by Hsu and Ramos [1] as a new class of embeddings, *Deconditional Mean Embeddings* (DME) are natural counterparts of CMEs. While CME $\mu_{X|Y=y} \in \mathcal{H}_k$ allows to take the conditional expectation of any $f \in \mathcal{H}_k$ through inner product $\mathbb{E}_X[f(X)|Y = y] = \langle f, \mu_{X|Y=y}\rangle_k$, the DME denoted $\mu_{X=x|Y} \in \mathcal{H}_\ell$ solves the inverse problem[3] and allows to recover the initial function of which the conditional expectation was taken, through inner product $\langle \mathbb{E}_X[f(X)|Y = \cdot], \mu_{X=x|Y}\rangle_\ell = f(x)$.

The associated Hilbert space operator, the *Deconditional Mean Operator* (DMO), is thus defined as the operator $D_{X|Y} : \mathcal{H}_k \to \mathcal{H}_\ell$ such that $D_{X|Y}^{\top}\mathbb{E}_X[f(X)|Y = \cdot] = f$, $\forall f \in \mathcal{H}_k$. It admits an expression in terms of CMO and cross-covariance operators $D_{X|Y} = (C_{X|Y}C_{YY})^{\top}(C_{X|Y}C_{YY}(C_{X|Y})^{\top})^{-1}$ provided $\ell_y \in \text{Range}(C_{YY})$ and $k_x \in \text{Range}(C_{X|Y}C_{YY}C_{X|Y}^{\top})$, $\forall y \in \mathcal{Y}$ and $\forall x \in \mathcal{X}$. A nonparametric empirical estimate of the DMO using datasets $\mathcal{D}_1$ and $\mathcal{D}_2$ as described above, is given by $\hat{D}_{X|Y} = \mathbf{\Psi}_{\tilde{\mathbf{y}}}(\mathbf{A}^{\top}\mathbf{K}_{\mathbf{xx}}\mathbf{A} + M\epsilon\mathbf{I}_M)^{-1}\mathbf{A}^{\top}\mathbf{\Phi}_{\mathbf{x}}^{\top}$ where $\epsilon > 0$ is a regularisation term and $\mathbf{A} := (\mathbf{L}_{\mathbf{yy}} + N\lambda\mathbf{I})^{-1}\mathbf{L}_{\mathbf{y}\tilde{\mathbf{y}}}$ can be interpreted as a mediation operator. Applying the DMO to expected responses $\tilde{\mathbf{z}}$, Hsu and Ramos [1] are able to recover an estimate of $f$ as

$$\hat{f}(x) = k(x, \mathbf{x})\mathbf{A}\left(\mathbf{A}^{\top}\mathbf{K}_{\mathbf{xx}}\mathbf{A} + M\epsilon\mathbf{I}_M\right)^{-1}\tilde{\mathbf{z}}. \tag{2}$$

Note that since separate samples $\tilde{\mathbf{y}}$ can be used to estimate $C_{YY}$, this naturally fits a mediating variables setup where $\mathbf{x}$ and the conditional means $\tilde{\mathbf{z}}$ are not jointly observed.

## 3 Deconditioning with Gaussian processes

In this section, we introduce *Conditional Mean Process* (CMP), a stochastic process stemming from the conditional expectation of a GP. We provide a characterisation of the CMP and show that the corresponding posterior mean of its integrand is a DME-based estimator. We also derive in Appendix B a variational formulation of our model that scales to large datasets and demonstrate its performance in Section 5.

For simplicity, we put aside observations bagging in Sections 3.1 and 3.2, our contributions being relevant to the general theory of DMEs. We return to a bagged formalism in Section 3.3 and extend deconditioning to the multiresolution setup inherent to the mediated statistical downscaling application. In what follows, $\mathcal{X}$ is a measurable space, $\mathcal{Y}$ a Borel space and feature maps $k_x$ and $\ell_y$ are Borel-measurable functions for any $x \in \mathcal{X}, y \in \mathcal{Y}$. All proofs and derivations of the paper are included in the appendix.

### 3.1 Conditional Mean Process

Bayesian quadrature [14, 31, 32] is based on the observation that the integral of a GP with respect to some marginal measure is a Gaussian random variable. We start by probing the implications of integrating with respect to conditional distribution $\mathbb{P}_{X|Y=y}$ and considering such integrals as functions of the conditioning variable $y$. This gives rise to the notion of conditional mean processes.

**Definition 3.1** (Conditional Mean Process). Let $f \sim \mathcal{GP}(m, k)$ with integrable sample paths, i.e. $\int_{\mathcal{X}} |f|\,\mathrm{d}\mathbb{P}_X < \infty$ a.s. The CMP induced by $f$ with respect to $\mathbb{P}_{X|Y}$ is defined as the stochastic

---

[3]the slightly unusual notation $\mu_{X=x|Y}$ is taken from Hsu and Ramos [1] and is meant to contrast the usual conditioning $\mu_{X|Y=y}$

process $\{g(y) \; : \; y \in \mathcal{Y}\}$ given by

$$g(y) = \mathbb{E}_X[f(X)|Y = y] = \int_{\mathcal{X}} f(x)\, \mathrm{d}\mathbb{P}_{X|Y=y}(x). \tag{3}$$

By linearity of the integral, it is clear that $g(y)$ is a Gaussian random variable for each $y \in \mathcal{Y}$. The sample paths integrability requirement ensures $g$ is well-defined almost everywhere. The following result characterizes CMP as a GP on $\mathcal{Y}$.

**Proposition 3.2** (CMP characterization). *Suppose $\mathbb{E}_X[|m(X)|] < \infty$ and $\mathbb{E}_X[\|k_X\|_k] < \infty$ and let $(X', Y') \sim \mathbb{P}_{XY}$. Then $g$ is a Gaussian process $g \sim \mathcal{GP}(\nu, q)$ a.s. , specified by*

$$\nu(y) = \mathbb{E}_X[m(X)|Y = y] \qquad q(y, y') = \mathbb{E}_{X,X'}[k(X, X')|Y = y, Y' = y'] \tag{4}$$

$\forall y, y' \in \mathcal{Y}$. *Furthermore,* $q(y, y') = \langle \mu_{X|Y=y}, \mu_{X|Y=y'} \rangle_k$ *a.s.*

Intuitively, the CMP can be understood as a GP on the conditional means where its covariance $q(y, y')$ is induced by the similarity between the CMEs at $y$ and $y'$. Resorting to the kernel $\ell$ defined on $\mathcal{Y}$, we can reexpress the covariance using Hilbert space operators as $q(y, y') = \langle C_{X|Y}\ell_y, C_{X|Y}\ell_{y'} \rangle_k$. A natural nonparametric estimate of the CMP covariance thus comes using the CMO estimator from (1) as $\hat{q}(y, y') = \ell(y, \mathbf{y})\left(\mathbf{L_{yy}} + N\lambda\mathbf{I}_N\right)^{-1}\mathbf{K_{xx}}\left(\mathbf{L_{yy}} + N\lambda\mathbf{I}_N\right)^{-1}\ell(\mathbf{y}, y')$. When $m \in \mathcal{H}_k$, the mean function can be written as $\nu(y) = \langle \mu_{X|Y=y}, m \rangle_k$ for which we can also use empirical estimate $\hat{\nu}(y) = \ell(y, \mathbf{y})\left(\mathbf{L_{yy}} + N\lambda\mathbf{I}_N\right)^{-1}\mathbf{\Phi_x}^\top m$. Finally, one can also quantify the covariance between the CMP $g$ and its integrand $f$, i.e. $\mathrm{Cov}(f(x), g(y)) = \mathbb{E}_X[k(x, X)|Y = y]$. Under the same assumptions as Proposition 3.2, this covariance can be expressed using mean embeddings, i.e. $\mathrm{Cov}(f(x), g(y)) = \langle k_x, \mu_{X|Y=y} \rangle_k$ and admits empirical estimate $k(x, \mathbf{x})\left(\mathbf{L_{yy}} + N\lambda\mathbf{I}_N\right)^{-1}\ell(\mathbf{y}, y)$.

## 3.2 Deconditional Posterior

Given independent observations introduced above, $\mathcal{D}_1 = \{\mathbf{x}, \mathbf{y}\}$ and $\mathcal{D}_2 = \{\tilde{\mathbf{y}}, \tilde{\mathbf{z}}\}$, we may now consider an additive noise model with CMP prior on aggregate observations $\tilde{\mathbf{z}}|\tilde{\mathbf{y}} \sim \mathcal{N}(g(\tilde{\mathbf{y}}), \sigma^2\mathbf{I}_M)$. Let $\mathbf{Q_{\tilde{y}\tilde{y}}} := q(\tilde{\mathbf{y}}, \tilde{\mathbf{y}})$ be the kernel matrix induced by $q$ on $\tilde{\mathbf{y}}$ and let $\mathbf{\Upsilon} := \mathrm{Cov}(f(\mathbf{x}), \tilde{\mathbf{z}}) = \mathbf{\Phi_x}^\top C_{X|Y}\mathbf{\Psi_{\tilde{y}}}$ be the cross-covariance between $f(\mathbf{x})$ and $\tilde{\mathbf{z}}$. The joint normality of $f(\mathbf{x})$ and $\tilde{\mathbf{z}}$ gives

$$\begin{bmatrix} f(\mathbf{x}) \\ \tilde{\mathbf{z}} \end{bmatrix} \mid \mathbf{y}, \tilde{\mathbf{y}} \sim \mathcal{N}\left(\begin{bmatrix} m(\mathbf{x}) \\ \nu(\tilde{\mathbf{y}}) \end{bmatrix}, \begin{bmatrix} \mathbf{K_{xx}} & \mathbf{\Upsilon} \\ \mathbf{\Upsilon}^\top & \mathbf{Q_{\tilde{y}\tilde{y}}} + \sigma^2\mathbf{I}_M \end{bmatrix}\right). \tag{5}$$

Using Gaussian conditioning, we can then readily derive the posterior distribution of the underlying GP field $f$ given the aggregate observations $\tilde{\mathbf{z}}$ corresponding to $\tilde{\mathbf{y}}$. The latter can naturally be degenerated if observations are paired, i.e. $\mathbf{y} = \tilde{\mathbf{y}}$. This formulation can be seen as an example of the inter-domain GP [33], where we utilise the observed conditional means $\tilde{\mathbf{z}}$ as inter-domain inducing features for inference of $f$.

**Proposition 3.3** (Deconditional Posterior). *Given aggregate observations $\tilde{\mathbf{z}}$ with homoscedastic noise $\sigma^2$, the deconditional posterior of $f$ is defined as the Gaussian process $f|\tilde{\mathbf{z}} \sim \mathcal{GP}(m_{\mathrm{d}}, k_{\mathrm{d}})$ where*

$$m_{\mathrm{d}}(x) = m(x) + k_x^\top C_{X|Y}\mathbf{\Psi_{\tilde{y}}}(\mathbf{Q_{\tilde{y}\tilde{y}}} + \sigma^2\mathbf{I}_M)^{-1}(\tilde{\mathbf{z}} - \nu(\tilde{\mathbf{y}})), \tag{6}$$

$$k_{\mathrm{d}}(x, x') = k(x, x') - k_x^\top C_{X|Y}\mathbf{\Psi_{\tilde{y}}}(\mathbf{Q_{\tilde{y}\tilde{y}}} + \sigma^2\mathbf{I}_M)^{-1}\mathbf{\Psi_{\tilde{y}}}^\top C_{X|Y}^\top k_{x'}. \tag{7}$$

Substituting terms by their empirical forms, we can define a nonparametric estimate of the $m_{\mathrm{d}}$ as

$$\hat{m}_{\mathrm{d}}(x) := m(x) + k(x, \mathbf{x})\mathbf{A}(\hat{\mathbf{Q}}_{\tilde{\mathbf{y}}\tilde{\mathbf{y}}} + \sigma^2\mathbf{I}_M)^{-1}(\tilde{\mathbf{z}} - \hat{\nu}(\tilde{\mathbf{y}}))) \tag{8}$$

which, when $m = 0$, reduces to the DME-based estimator in (2) by taking the noise variance $\frac{\sigma^2}{N}$ as the inverse regularization parameter. Hsu and Ramos [1] recover a similar posterior mean expression in their Bayesian interpretation of DME. However, they do not link the distributions of $f$ and its CMP, which leads to much more complicated chained inference derivations combining fully Bayesian and MAP estimates, while we naturally recover it using simple Gaussian conditioning.

Likewise, an empirical estimate of the deconditional covariance is given by

$$\hat{k}_{\mathrm{d}}(x, x') := k(x, x') - k(x, \mathbf{x})\mathbf{A}(\hat{\mathbf{Q}}_{\tilde{\mathbf{y}}\tilde{\mathbf{y}}} + \sigma^2\mathbf{I}_M)^{-1}\mathbf{A}^\top k(\mathbf{x}, x'). \tag{9}$$

Interestingly, the latter can be rearranged to write as the difference between the original kernel and the kernel undergoing conditioning and deconditioning steps $\hat{k}_{\mathrm{d}}(x, x') = k(x, x') - \langle k_x, \hat{D}_{X|Y}\hat{C}_{X|Y}k_{x'}\rangle_k$. This can be interpreted as a measure of reconstruction quality, which degenerates in the case of perfect deconditioning, i.e. $\hat{D}_{X|Y}\hat{C}_{X|Y} = \mathrm{Id}_{\mathcal{H}_k}$.

### 3.3 Deconditioning and multiresolution data

Downscaling application would typically correspond to multiresolution data, with bag dataset $^b\mathcal{D}_1 = \{(^b\boldsymbol{x}_j, y_j)\}_{j=1}^N$ where $^b\boldsymbol{x}_j = \{x_j^{(i)}\}_{i=1}^{n_j}$. In this setup, the mismatch in size between vector concatenations $\mathbf{x} = [x_1^{(1)} \ \ldots \ x_N^{(n_N)}]^\top$ and $\mathbf{y} = [y_1 \ \ldots \ y_N]^\top$ prevents from readily applying (1) to estimate the CMO and thus infer the deconditional posterior. There is, however, a straightforward approach to alleviate this: simply replicate bag-level covariates $y_j$ to match bags sizes $n_j$. Although simple, this method incurs a $\mathcal{O}((\sum_{j=1}^N n_j)^3)$ cost due to matrix inversion in (1).

Alternatively, since bags $^b\boldsymbol{x}_j$ are sampled from conditional distribution $\mathbb{P}_{X|Y=y_j}$, unbiased Monte Carlo estimators of CMEs are given by $\hat{\mu}_{X|Y=y_j} = \frac{1}{n_j}\sum_{i=1}^{n_j} k_{x_j^{(i)}}$. Let $\hat{\mathbf{M}}_{\mathbf{y}} = [\hat{\mu}_{X|Y=y_1}\ldots\hat{\mu}_{X|Y=y_N}]^\top$ denote their concatenation along columns. We can then rewrite the cross-covariance operator as $C_{XY} = \mathbb{E}_Y[\mathbb{E}_X[k_X|Y]\otimes\ell_Y]$ and hence take $\frac{1}{N}\hat{\mathbf{M}}_{\mathbf{y}}\boldsymbol{\Psi}_{\mathbf{y}}^\top$ as an estimate for $C_{XY}$. Substituting empirical forms into $C_{X|Y} = C_{XY}C_{YY}^{-1}$ and applying Woodbury identity, we obtain an alternative CMO estimator that only requires inversion of a $N \times N$ matrix. We call it *Conditional Mean Shrinkage Operator* and define it as

$$^S\hat{C}_{X|Y} := \hat{\mathbf{M}}_{\mathbf{y}}(\mathbf{L}_{\mathbf{yy}} + \lambda N\mathbf{I}_N)^{-1}\boldsymbol{\Psi}_{\mathbf{y}}^\top. \tag{10}$$

This estimator can be seen as a generalisation of the Kernel Mean Shrinkage Estimator [34] to the conditional case. We provide in Appendix C modifications of (8) and (9) including this estimator for the computation of the deconditional posterior.

## 4 Deconditioning as regression

In Section 3.2, we obtain a DMO-based estimate for the posterior mean of $f|\tilde{\mathbf{z}}$. When the estimate gets closer to the exact operator, the uncertainty collapses and the Bayesian view meets the frequentist. It is however unclear how the empirical operators effectively converge in finite data size. Adopting an alternative perspective, we now demonstrate that the DMO estimate can be obtained as the minimiser of a two-staged vector-valued regression. This frequentist turn enables us to leverage rich theory of vector-valued regression and establish under mild assumptions a convergence rate on the DMO estimator, with conditions to fulfill minimax optimality in terms of statistical-computational efficiency. In the following, we briefly review CMO's vector-valued regression viewpoint and construct an analogous regression problem for DMO. We refer the reader to [35] for a comprehensive overview of vector-valued RKHS theory. As for Sections 3.1 and 3.2, this section contributes to the general theory of DMEs and we hence put aside the bag notations.

**Stage 1: Regressing the Conditional Mean Operator** As first introduced by Grünewälder et al. [23] and generalised to infinite dimensional spaces by Singh et al. [25], estimating $C_{X|Y}^\top$ is equivalent to solving a vector-valued kernel ridge regression problem in the hypothesis space of Hilbert-Schmidt operators from $\mathcal{H}_k$ to $\mathcal{H}_\ell$, denoted as $\mathrm{HS}(\mathcal{H}_k, \mathcal{H}_\ell)$. Specifically, we may consider the operator-valued kernel defined over $\mathcal{H}_k$ as $\Gamma(f, f') := \langle f, f'\rangle_k\,\mathrm{Id}_{\mathcal{H}_\ell}$. We denote $\mathcal{H}_\Gamma$ the $\mathcal{H}_\ell$-valued RKHS spanned by $\Gamma$ with norm $\|\cdot\|_\Gamma$, which can be identified to $\mathrm{HS}(\mathcal{H}_k, \mathcal{H}_\ell)$[4]. Singh et al. [25] frame CMO regression as the minimisation surrogate discrepancy $\mathcal{E}_{\mathrm{c}}(C) := \mathbb{E}_{X,Y}\left[\|k_X - C^\top\ell_Y\|_k^2\right]$, to which they substitute an empirical regularised version restricted to $\mathcal{H}_\Gamma$ given by

$$\hat{\mathcal{E}}_{\mathrm{c}}(C) := \frac{1}{N}\sum_{i=1}^N \|k_{x_i} - C^\top\ell_{y_i}\|_k^2 + \lambda\|C\|_\Gamma^2 \qquad C \in \mathcal{H}_\Gamma \qquad \lambda > 0 \tag{11}$$

This $\mathcal{H}_k$-valued kernel ridge regression problem admits a closed-form minimiser which shares the same empirical form as the CMO, i.e. $\hat{C}_{X|Y}^\top = \arg\min_{C\in\mathcal{H}_\Gamma}\hat{\mathcal{E}}_{\mathrm{c}}(C)$ [23, 25].

---

[4] $\mathcal{H}_\Gamma = \overline{\mathrm{Span}}\{\Gamma_f h, f \in \mathcal{H}_k, h \in \mathcal{H}_\ell\} = \overline{\mathrm{Span}}\{f \otimes h, f \in \mathcal{H}_k, h \in \mathcal{H}_\ell\} = \overline{\mathcal{H}_k \otimes \mathcal{H}_\ell} \cong \mathrm{HS}(\mathcal{H}_k, \mathcal{H}_\ell)$

**Stage 2 : Regressing the Deconditional Mean Operator** The DMO on the other hand is defined as the operator $D_{X|Y} : \mathcal{H}_k \to \mathcal{H}_\ell$ such that $\forall f \in \mathcal{H}_k$, $D_{X|Y} C_{X|Y}^\top f = f$. Since deconditioning corresponds to finding a pseudo-inverse to the CMO, it is natural to consider a reconstruction objective $\mathcal{E}_d(D) := \mathbb{E}_Y\left[\|\ell_Y - DC_{X|Y}\ell_Y\|_\ell^2\right]$. Introducing a novel characterization of the DMO, we propose to minimise this objective in the hypothesis space of Hilbert-Schmidt operators $\mathrm{HS}(\mathcal{H}_k, \mathcal{H}_\ell)$ which identifies to $\mathcal{H}_\Gamma$. As per above, we denote $\hat{C}_{X|Y}$ the empirical CMO learnt in Stage 1, and we substitute the loss with an empirical regularised formulation on $\mathcal{H}_\Gamma$

$$\hat{\mathcal{E}}_d(D) := \frac{1}{M}\sum_{j=1}^M \|\ell_{\tilde{y}_j} - D\hat{C}_{X|Y}\ell_{\tilde{y}_j}\|_\ell^2 + \epsilon\|D\|_\Gamma^2 \qquad D \in \mathcal{H}_\Gamma \qquad \epsilon > 0 \qquad (12)$$

**Proposition 4.1** (Empirical DMO as vector-valued regressor). *The minimiser of the empirical reconstruction risk is the empirical DMO, i.e. $\hat{D}_{X|Y} = \arg\min_{D\in\mathcal{H}_\Gamma} \hat{\mathcal{E}}_d(D)$*

Since it requires to estimate the CMO first, minimising (12) can be viewed as a two-staged vector value regression problem.

**Convergence results** Following the footsteps of [24, 25], this perspective enables us to state the performance of the DMO estimate in terms of asymptotic convergence of the objective $\mathcal{E}_d$. As in Caponnetto and De Vito [36], we must restrict the class of probability measure for $\mathbb{P}_{XY}$ and $\mathbb{P}_Y$ to ensure uniform convergence even when $\mathcal{H}_k$ is infinite dimensional. The family of distribution considered is a general class of priors that does not assume parametric distributions and is parametrized by two variables: $b > 1$ controls the effective input dimension and $c \in ]1, 2]$ controls functional smoothness. Mild regularity assumptions are also placed on the original spaces $\mathcal{X}, \mathcal{Y}$, their corresponding RKHS $\mathcal{H}_k, \mathcal{H}_\ell$ and the vector-valued RKHS $\mathcal{H}_\Gamma$. We discuss these assumptions in details in Appendix D. Importantly, while $\mathcal{H}_k$ can be infinite dimensional, we nonetheless have to assume the RKHS $\mathcal{H}_\ell$ is finite dimensional. In further research, we will relax this assumption.

**Theorem 4.2** (Empirical DMO Convergence Rate). *Denote $D_{\mathbb{P}_Y} = \arg\min_{D\in\mathcal{H}_\Gamma} \mathcal{E}_d(D)$. Assume assumptions stated in Appendix D are satisfied. In particular, let $(b, c)$ and $(0, c')$ be the parameters of the restricted class of distribution for $\mathbb{P}_Y$ and $\mathbb{P}_{XY}$ respectively and let $\iota \in ]0, 1]$ be the Hölder continuity exponent in $\mathcal{H}_\Gamma$. Then, if we choose $\lambda = N^{-\frac{1}{c'+1}}$, $N = M^{\frac{a(c'+1)}{\iota(c'-1)}}$ where $a > 0$, we have the following result,*

- *If $a \le \frac{b(c+1)}{bc+1}$, then $\mathcal{E}_d(\hat{D}_{X|Y}) - \mathcal{E}_d(D_{\mathbb{P}_Y}) = \mathcal{O}(M^{\frac{-ac}{c+1}})$ with $\epsilon = M^{\frac{-a}{c+1}}$*

- *If $a \ge \frac{b(c+1)}{bc+1}$, then $\mathcal{E}_d(\hat{D}_{X|Y}) - \mathcal{E}_d(D_{\mathbb{P}_Y}) = \mathcal{O}(M^{\frac{-bc}{bc+1}})$ with $\epsilon = M^{\frac{-b}{bc+1}}$*

This theorem underlines a trade-off between the computational and statistical efficiency with respect to the datasets cardinalities $N = |\mathcal{D}_1|$, $M = |\mathcal{D}_2|$ and the problem difficulty $b, c, c'$. For $a \le \frac{b(c+1)}{bc+1}$, smaller $a$ means less samples from $\mathcal{D}_1$ at fixed $M$ and thus computational savings. But it also hampers convergence, resulting in reduced statistical efficiency. At $a = \frac{b(c+1)}{bc+1} < 2$, convergence rate is a minimax computational-statistical efficiency optimal, i.e. convergence rate is optimal with smallest possible $M$. We note that at this optimal, $N > M$ and which means less samples are required from $\mathcal{D}_2$. $a \ge \frac{b(c+1)}{bc+1}$ does not improve the convergence rate but only increases the size of $\mathcal{D}_1$ and hence the computational cost it bears.

## 5 Deconditional Downscaling Experiments

We demonstrate and evaluate our CMP-based downscaling approaches on both synthetic experiments and a challenging atmospheric temperature field downscaling problem with unmatched multi-resolution data. We denote the exact CMP deconditional posterior from Section 3 as CMP, the CMP using with efficient shrinkage CMO estimation as S-CMP and the variational formulation as VARCMP. They are compared against VBAGG [9] — which we describe below — and a GP regression [14] baseline (GPR) modified to take bags centroids as the input. Experiments are implemented in *PyTorch* [37, 38], all code and datasets are made available[5] and computational details are provided in Appendix E.

---

[5] https://github.com/shahineb/deconditional-downscaling

**Variational Aggregate Learning**   VBAGG is introduced by Law et al. [9] as a variational aggregate learning framework to disaggregate exponential family models, with emphasis on the Poisson family. We consider its application to the Gaussian family, which models the relationship between aggregate targets $z_j$ and bag covariates $\{x_j^{(i)}\}_i$ by bag-wise averaging of a GP prior on the function of interest. In fact, the Gaussian VBAGG corresponds exactly to a special case of CMP on matched data, where the bag covariates are simply one hot encoded indices with kernel $\ell(j, j') = \delta(j, j')$ where $\delta$ is the Kronecker delta. However, VBAGG cannot handle unmatched data as bag indices do not instill the smoothness that is used for mediation. For fair analysis, we compare variational methods VARCMP and VBAGG together, and exact methods CMP/S-CMP to an exact version of VBAGG, which we implement and refer to as BAGG-GP.

## 5.1   Swiss Roll

The *scikit-learn* [39] swiss roll manifold sampling function allows to generate a 3D manifold of points $x \in \mathbb{R}^3$ mapped with their position along the manifold $t \in \mathbb{R}$. Our objective will be to recover $t$ for each point $x$ by only observing $t$ at an aggregate level. In the first experiment, we compare our model to existing weakly supervised spatial disaggregation methods when all high-resolution covariates are matched with a coarse-level aggregate target. We then proceed to withdraw this requirement in a companion experiment.

### 5.1.1   Direct matching

**Experimental Setup**   Inspired by the experimental setup from Law et al. [9], we regularly split space along height $B - 1$ times as depicted in Figure 2 and group together manifold points within each height level, hence mixing together points with very different positions on the manifold. We obtain bags of samples $\{({}^b\boldsymbol{x}_j, \boldsymbol{t}_j)\}_{j=1}^B$ where the $j^{\text{th}}$ bag contains $n_j$ points ${}^b\boldsymbol{x}_j = \{x_j^{(i)}\}_{i=1}^{n_j}$ and their corresponding targets $\boldsymbol{t}_j = \{t_j^{(i)}\}_{i=1}^{n_j}$. We then construct bag aggregate targets by taking noisy bag targets average $z_j := \frac{1}{n_j} \sum_{i=1}^{n_j} t_j^{(i)} + \varepsilon_j$, where $\varepsilon_j \sim \mathcal{N}(0, \sigma^2)$. We thus obtain matched weakly supervised bag learning dataset $\mathcal{D}^\circ = \{({}^b\boldsymbol{x}_j, z_j)\}_{j=1}^B$. Since each bag corresponds to a height-level, the center altitude of each height split $y_j \in \mathbb{R}$ is a natural candidate bag-level covariate that informs on relative positions of the bags. We can

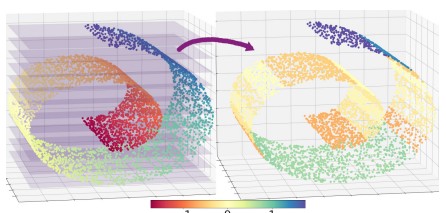

Figure 2: **Step 1:** Split space regularly along height. **Step 2:** Group points into height-level bags. **Step 3:** Average points targets into bag-level aggregate targets.

augment the above dataset as $\mathcal{D} = \{({}^b\boldsymbol{x}_j, y_j, z_j)\}_{j=1}^B$. Using these bag datasets, we now wish to downscale aggregate targets $z_j$ to recover the unobserved manifold locations $\{\boldsymbol{t}_j\}_{j=1}^B$ and be able to query the target at any previously unseen input $x$.

**Models**   We use a zero-mean prior on $f$ and choose a Gaussian kernel for $k$ and $\ell$. Inducing points location is initialized with K-means++ procedure for VARCMP and VBAGG such that they spread evenly across the manifold. For exact methods, kernel hyperparameters and noise variance $\sigma^2$ are learnt on $\mathcal{D}$ by optimising the marginal likelihood. For VARCMP, they are learnt jointly with variational distribution parameters by maximising an evidence lower bound objective. While CMP-based methods can leverage bag-level covariates $y_j$, baselines are restricted to learn from $\mathcal{D}^\circ$. Adam optimiser [40] is used in all experiments.

**Results**   We test models against unobserved groundtruth $\{\boldsymbol{t}_j\}_{j=1}^B$ by evaluating the root mean square error (RMSE) to the posterior mean. Table 1 shows that CMP, S-CMP and VARCMP outperform their corresponding counterparts i.e. BAGG-GP and VBAGG, with statistical significance confirmed by a Wilcoxon signed-rank test in Appendix E. Most notably, this shows that the additional knowledge on bag-level dependence instilled by $\ell$ is reflected even in a setting where each bag is matched with an aggregate target.

Table 1: RMSE of the swissroll experiment for models trained over directly and indirectly matched datasets ; scores averaged over 20 seeds and 1 s.d is reported ; * indicates our proposed methods.

| Matching | CMP* | S-CMP* | VARCMP* | BAGG-GP | VBAGG | GPR |
|---|---|---|---|---|---|---|
| Direct | $0.33_{\pm 0.06}$ | $0.25_{\pm 0.04}$ | $\mathbf{0.18}_{\pm 0.04}$ | $0.60_{\pm 0.01}$ | $0.22_{\pm 0.04}$ | $0.70_{\pm 0.05}$ |
| Indirect | $\mathbf{0.80}_{\pm 0.14}$ | $1.05_{\pm 0.04}$ | $0.87_{\pm 0.07}$ | $1.13_{\pm 0.11}$ | $1.46_{\pm 0.34}$ | $1.04_{\pm 0.05}$ |

### 5.1.2 Indirect matching

**Experimental Setup** We now impose indirect matching through mediating variable $y_j$. We randomly select $N = \lfloor \frac{B}{2} \rfloor$ bags which we consider to be the $N$ first ones without loss of generality and split $\mathcal{D}$ into $\mathcal{D}_1 = \{(^b x_j, y_j)\}_{j=1}^N$ and $\mathcal{D}_2 = \{(\tilde{y}_j, \tilde{z}_j)\}_{j=1}^{B-N} = \{(y_{N+j}, z_{N+j})\}_{j=1}^{B-N}$, such that no pair of covariates bag $^b x_j$ and aggregate target $\tilde{z}_j$ are jointly observed.

**Models** CMP-based methods are naturally able to learn from this setting and are trained by independently drawing samples from $\mathcal{D}_1$ and $\mathcal{D}_2$. Baseline methods however require bags of covariates to be matched with an aggregate bag target. To remedy this, we place a separate prior $g \sim \mathcal{GP}(0, \ell)$ and fit regression model $\tilde{z}_j = g(\tilde{y}_j) + \varepsilon_j$ over $\mathcal{D}_2$. We then use the predictive posterior mean to augment the first dataset as $\mathcal{D}_1' = \left\{ \left(^b x_j, \mathbb{E}[g(y_j)|\mathcal{D}_2]\right) \right\}_{j=1}^N$. This dataset can then be used to train BAGG-GP, VBAGG and GPR.

**Results** For comparison, we use the same evaluation as in the direct matching experiment. Table 1 underlines an anticipated drop in RMSE for all models, but we observe that BAGG-GP and VBAGG suffer most from the mediated matching of the dataset while CMP and VARCMP report best scores by a substantial margin. This highlights how using a separate prior on $g$ to mediate $\mathcal{D}_1$ and $\mathcal{D}_2$ turns out to be suboptimal in contrast to using the prior naturally implied by CMP. While it is more computationally efficient than CMP, we observe a relative drop in performance for S-CMP.

### 5.2 Mediated downscaling of atmospheric temperature

Given the large diversity of sources and formats of remote sensing and model data, expecting directly matched observations is often unrealistic [41]. For example, two distinct satellite products will often provide low and high resolution imagery that can be matched neither spatially nor temporally [2, 3, 4, 5]. Climate simulations [19, 20, 21] on the other hand provide a comprehensive coarse resolution coverage of meteorological variables that can serve as a mediating dataset.

For the purpose of demonstration, we create an experimental setup inspired by this problem using Coupled Model Intercomparison Project Phase 6 (CMIP6) [19] simulation data. This grants us access to groundtruth high-resolution covariates to facilitate model evaluation.

**Experimental Setup** We collect monthly mean 2D atmospheric fields simulation from CMIP6 data [42, 43] for the following variables: air temperature at cloud top (T), mean cloud top pressure (P), total cloud cover (TCC) and cloud albedo ($\alpha$). First, we collocate TCC and $\alpha$ onto a HR latitude-longitude grid of size $360 \times 720$ to obtain fine grained fields (latitude$^{\mathrm{HR}}$, longitude$^{\mathrm{HR}}$, altitude$^{\mathrm{HR}}$, TCC$^{\mathrm{HR}}$, $\alpha^{\mathrm{HR}}$) augmented with a static HR surface altitude field. Then we collocate P and T onto a LR grid of size $21 \times 42$ to obtain coarse grained fields (latitude$^{\mathrm{LR}}$, longitude$^{\mathrm{LR}}$, P$^{\mathrm{LR}}$, T$^{\mathrm{LR}}$). We denote by $B$ the number of low resolution pixels.

Our objective is to disaggregate T$^{\mathrm{LR}}$ to the HR fields granularity. We assimilate the $j^{\mathrm{th}}$ coarse temperature pixel to an aggregate target $z_j := \mathrm{T}_j^{\mathrm{LR}}$ corresponding to bag $j$. Each bag includes HR covariates $^b x_j = \{x_j^{(i)}\}_{i=1}^{n_j} := \{(\mathrm{latitude}_j^{\mathrm{HR}(i)}, \mathrm{longitude}_j^{\mathrm{HR}(i)}, \mathrm{altitude}_j^{\mathrm{HR}(i)}, \mathrm{TCC}_j^{\mathrm{HR}(i)}, \alpha_j^{\mathrm{HR}(i)})\}_{i=1}^{n_j}$. To emulate unmatched observations, we randomly select $N = \lfloor \frac{B}{2} \rfloor$ of the bags $\{^b x_j\}_{j=1}^N$ and keep the opposite half of LR observations $\{z_{N+j}\}_{j=1}^{B-N}$, such that there is no single aggregate bag target that corresponds to one of the available bags. Finally, we choose the pressure field P$^{\mathrm{LR}}$ as the mediating variable. We hence compose a third party low resolution field of bag-level covariates $y_j := (\mathrm{latitude}_j^{\mathrm{LR}}, \mathrm{longitude}_j^{\mathrm{LR}}, \mathrm{P}_j^{\mathrm{LR}})$ which can separately be matched with both above sets to obtain datasets $\mathcal{D}_1 = \{(x_j, y_j)\}_{j=1}^N$ and $\mathcal{D}_2 = \{(\tilde{y}_j, \tilde{z}_j)\}_{j=1}^{B-N} = \{(y_{N+j}, z_{N+j})\}_{j=1}^{B-N}$.

Figure 3: **Left:** High-resolution atmoshperic covariates used for prediction; **Center-Left:** Observed low-resolution temperature field, grey pixels are unobserved; **Center** Unobserved high-resolution groundtruth temperature field; **Center-Right:** VARCMP deconditional posterior mean; **Right** 95% confidence region size on prediction; temperature values are in Kelvin.

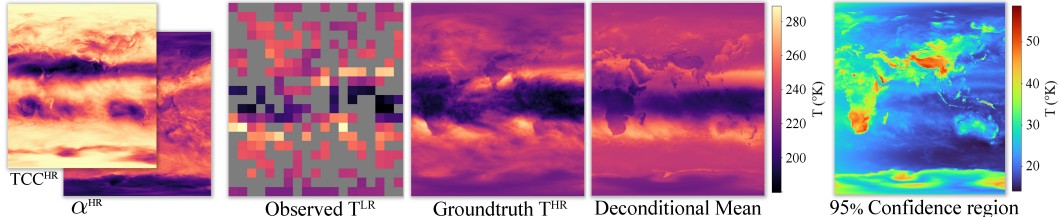

Table 2: Downscaling similarity scores of posterior mean against groundtruth high resolution cloud top temperature field ; averaged over 10 seeds; we report 1 s.d. ; "↓": lower is better ; "↑": higher is better.

| Model | RMSE ↓ | MAE ↓ | Corr. ↑ | SSIM ↑ |
|---|---|---|---|---|
| VARGPR | $8.02_{\pm 0.28}$ | $5.55_{\pm 0.17}$ | $0.831_{\pm 0.012}$ | $\mathbf{0.212}_{\pm 0.011}$ |
| VBAGG | $8.25_{\pm 0.15}$ | $5.82_{\pm 0.11}$ | $0.821_{\pm 0.006}$ | $0.182_{\pm 0.004}$ |
| VARCMP | $\mathbf{7.40}_{\pm 0.25}$ | $\mathbf{5.34}_{\pm 0.22}$ | $\mathbf{0.848}_{\pm 0.011}$ | $\mathbf{0.212}_{\pm 0.013}$ |

**Models Setup**   We only consider variational methods to scale to large number of pixels. VARCMP is naturally able to learn from indirectly matched data. We use a Matérn-1.5 kernel for rough spatial covariates (latitude, longitude) and a Gaussian kernel for atmospheric covariates (P, TCC, $\alpha$) and surface altitude. $k$ and $\ell$ are both taken as sums of Matérn and Gaussian kernels, and their hyperparameters are learnt along with noise variance during training. A high-resolution noise term is also introduced, with details provided in Appendix E. Inducing points locations are uniformly initialized across the HR grid. We replace GPR with an inducing point variational counterpart VARGPR [15]. Since baseline methods require a matched dataset, we proceed as with the unmatched swiss roll experiment and fit a GP regression model $g$ with kernel $\ell$ on $\mathcal{D}_2$ and then use its predictive posterior mean to obtain pseudo-targets for the bags of HR covariates from $\mathcal{D}_1$.

**Results**   Performance is evaluated by comparing downscaling deconditional posterior mean against original high resolution field $T^{HR}$ available in CMIP6 data [43], which we emphasise is never observed. We use random Fourier features [44] approximation of kernel $k$ to scale kernel evaluation to the HR covariates grid during testing. As reported in Table 2, VARCMP substantially outperforms both baselines with statistical significance provided in Appendix E. Figure 3 shows the reconstructed image with VARCMP, plots for other methods are included in the Appendix E. The model resolves statistical patterns from HR covariates into coarse resolution temperature pixels, henceforth reconstructing a faithful HR version of the temperature field.

## 6   Discussion

We introduced a scalable Bayesian solution to the mediated statistical downscaling problem, which handles unmatched multi-resolution data. The proposed approach combines Gaussian Processes with the framework of deconditioning using RKHSs and recovers previous approaches as its special cases. We provided convergence rates for the associated deconditioning operator. Finally, we demonstrated the advantages over spatial disaggregation baselines in synthetic data and in a challenging atmospheric fields downscaling problem.

In future work, exploring theoretical guarantees of the computationally efficient shrinkage formulation in a multi-resolution setting and relaxing finite dimensionality assumptions for the convergence rate will have fruitful practical and theoretical implications. Further directions also open up to quantify uncertainty over the deconditional posterior since it is computed using empirical estimates of the CMP covariance. This may be problematic if the mediating variable undergoes covariate shift between the two datasets.

## Acknowledgments

SLC is supported by the EPSRC and MRC through the OxWaSP CDT programme EP/L016710/1. SB is supported by the EU's Horizon 2020 research and innovation programme under Marie Skłodowska-Curie grant agreement No 860100. DS is supported in partly by Tencent AI lab and partly by the Alan Turing Institute (EP/N510129/1).

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
