**Supplementary materials for
"Deconditional Downscaling with Gaussian Processes"**

# A Proofs

## A.1 Proofs of Section 3

**Proposition A.1.** *Let $h : \mathcal{X} \to \mathbb{R}$ such that $\mathbb{E}[|h(X)|] < \infty$. Then, $\{y \in \mathcal{Y} \mid \mathbb{E}[|h(X)||Y = y] < \infty\}$ is a full measure set with respect to $\mathbb{P}_Y$.*

*Proof.* Since $\mathcal{X}$ is a Borel space and $\mathcal{Y}$ is measurable, the existence of a $\mathbb{P}_Y$-a.e. regular conditional probability distribution is guaranteed by [45, Theorem 6.3]. Now suppose $\mathbb{E}[|h(X)|] < \infty$ and let $\mathcal{Y}^o = \{y \in \mathcal{Y} \mid \mathbb{E}[|h(X)||Y = y] < \infty\}$. Since $\mathbb{E}[|h(X)|] = \mathbb{E}\left[\mathbb{E}[|h(X)| \mid Y]\right]$, the conditional expectation $\mathbb{E}[|h(X)| \mid Y]$ must have finite expectation almost everywhere, i.e. $\mathbb{P}_Y(\mathcal{Y}^o) = 1$. $\qquad\square$

**Proposition 3.2.** *Suppose $\mathbb{E}[|m(X)|] < \infty$ and $\mathbb{E}[\|k_X\|_k] < \infty$ and let $(X', Y') \sim \mathbb{P}_{XY}$. Then $g$ is a Gaussian process $g \sim \mathcal{GP}(\nu, q)$ a.s. , specified by*

$$\nu(y) = \mathbb{E}[m(X)|Y = y] \qquad q(y, y') = \mathbb{E}[k(X, X')|Y = y, Y' = y'] \tag{13}$$

*$\forall y, y' \in \mathcal{Y}$. Furthermore, $q(y, y') = \langle \mu_{X|Y=y}, \mu_{X|Y=y'} \rangle_k$ a.s.*

*Proof of Proposition 3.2.* We will assume for the sake of simplicity that $m = 0$ in the following derivations and will return to the case of an uncentered GP at the end of the proof.

**Show that $g(y)$ is in a space of Gaussian random variables** Let $(\Omega, \mathcal{F}, \mathbb{P})$ denote a probability space and $L^2(\Omega, \mathbb{P})$ the space of square integrable random variables endowed with standard inner product. $\forall x \in \mathcal{X}$, since $f(x)$ is Gaussian, then $f(x) \in L^2(\Omega, \mathbb{P})$. We can hence define $\mathcal{S}(f)$ as the closure in $L^2(\Omega, \mathbb{P})$ of the vector space spanned by $f$, i.e. $\mathcal{S}(f) := \overline{\text{Span}} \{f(x) : x \in \mathcal{X}\}$.

Elements of $\mathcal{S}(f)$ write as limits of centered Gaussian random variables, hence when their covariance sequence converge, they are normally distributed. Let $T \in \mathcal{S}(f)^\perp$, then we have $\mathbb{E}[Tf(x)] = 0$. Let $y \in \mathcal{Y}$, we also have

$$\mathbb{E}[Tg(y)] = \mathbb{E}\left[\int_{\mathcal{X}} Tf(x)\, d\mathbb{P}_{X|Y=y}\right] \tag{14}$$

In order to switch orders of integration, we need to show that the double integral satisfies absolute convergence.

$$\int_{\mathcal{X}} \mathbb{E}[|Tf(x)|]\, d\mathbb{P}_{X|Y=y}(x) \leq \int_{\mathcal{X}} \sqrt{\mathbb{E}[T^2]\mathbb{E}[f(x)^2]}\, d\mathbb{P}_{X|Y=y}(x) \tag{15}$$

$$= \sqrt{\mathbb{E}[T^2]} \int_{\mathcal{X}} \|k_x\|_k\, d\mathbb{P}_{X|Y=y}(x) \tag{16}$$

$$= \sqrt{\mathbb{E}[T^2]}\mathbb{E}[\|k_X\|_k|Y = y] \tag{17}$$

Since $T \in L^2(\Omega, \mathbb{P})$, $\mathbb{E}[T^2] < \infty$. Plus, as we assume that $\mathbb{E}[\|k_X\|_k] < \infty$, Proposition A.1 gives that $\mathbb{E}[\|k_X\|_k|Y = y] < \infty$ a.s. We can thus apply Fubini's theorem and obtain

$$\mathbb{E}[Tg(y)] = \int_{\mathcal{X}} \mathbb{E}[Tf(x)]\, d\mathbb{P}_{X|Y=y}(x) = 0 \text{ a.s.} \tag{18}$$

As this holds for any $T \in \mathcal{S}(f)^\perp$, we conclude that $g(y) \in \left(\mathcal{S}(f)^\perp\right)^\perp$ a.s. $\Rightarrow g(y) \in \mathcal{S}(f)$ a.s.. We cannot claim yet though that $g(y)$ is Gaussian since we do not know whether it results from a sequence of Gaussian variables with converging variance sequence. We now have to prove that $g(y)$ has a finite variance.

**Show that $g(y)$ has finite variance** We proceed by computing the expression of the covariance between $g(y)$ and $g(y')$ which is more general and yields the variance.

Let $y, y' \in \mathcal{Y}$, the covariance of $g(y)$ and $g(y')$ is given by

$$q(y, y') = \mathbb{E}[g(y)g(y')] - \mathbb{E}[g(y)]\mathbb{E}[g(y')] \tag{19}$$

$$= \mathbb{E}\left[\int_{\mathcal{X}} \int_{\mathcal{X}} f(x)f(x')\, d\mathbb{P}_{X|Y=y}(x)\, d\mathbb{P}_{X|Y=y'}(x')\right] \tag{20}$$

$$- \mathbb{E}\left[\int_{\mathcal{X}} f(x)\, d\mathbb{P}_{X|Y=y}(x)\right] \mathbb{E}\left[\int_{\mathcal{X}} f(x')\, d\mathbb{P}_{X|Y=y'}(x')\right] \tag{21}$$

Choosing $T$ as a constant random variable in the above, we can show that $\int_{\mathcal{X}} \mathbb{E}[|f(x)|]\, d\mathbb{P}_{X|Y=y}(x) < \infty$ a.s. We can hence apply Fubini's theorem to switch integration order in the mean terms (21) and obtain that $\mathbb{E}[g(y)] = 0$ since $f$ is centered.

To apply Fubini's theorem to (20), we need to show that the triple integration absolutely converges. Let $x, x' \in \mathcal{X}$, we know that $\mathbb{E}[|f(x)f(x')|] \leq \sqrt{\mathbb{E}[f(x)^2]\mathbb{E}[f(x')^2]} = \|k_x\|_k \|k_{x'}\|_k$. Using similar arguments as above, we obtain

$$\int_{\mathcal{X}} \int_{\mathcal{X}} \mathbb{E}[|f(x)f(x')|]\, d\mathbb{P}_{X|Y=y}(x)\, d\mathbb{P}_{X|Y=y'}(x') \leq \mathbb{E}[\|k_X\|_k | Y = y]\mathbb{E}[\|k_X\|_k | Y = y'] < \infty \text{ a.s.} \tag{22}$$

We can thus apply Fubini's theorem which yields

$$q(y, y') = \int_{\mathcal{X}} \int_{\mathcal{X}} \mathbb{E}[f(x)f(x')]\, d\mathbb{P}_{X|Y=y}(x)\, d\mathbb{P}_{X|Y=y'}(x') \tag{23}$$

$$= \int_{\mathcal{X}} \int_{\mathcal{X}} \underbrace{\mathrm{Cov}(f(x), f(x'))}_{k(x,x')}\, d\mathbb{P}_{X|Y=y}(x)\, d\mathbb{P}_{X|Y=y'}(x') \tag{24}$$

$$= \mathbb{E}[k(X, X') | Y = y, Y' = y'] \tag{25}$$

$$\leq \mathbb{E}[\|k_X\|_k | Y = y]\mathbb{E}[\|k_X\|_k | Y = y'] < \infty \text{ a.s.} \tag{26}$$

where $(X', Y')$ denote random variables with same joint distribution than $(X, Y)$ as defined in the proposition.

$g(y) \in \mathcal{S}(f)$ and has finite variance $q(y, y)$ a.s., it is thus a centered Gaussian random variable a.s. Furthermore, as this holds for any $y \in \mathcal{Y}$, then any finite subset of $\{g(y) : y \in \mathcal{Y}\}$ follows a multivariate normal distribution which shows that $g$ is a centered Gaussian process on $\mathcal{Y}$ and its covariance function is specified by $q$.

**Uncentered case $m \neq 0$** We now return to an uncentered GP prior on $f$ with assumption that $\mathbb{E}[|m(X)|] < \infty$. By Proposition A.1, we get that $\mathbb{E}[|m(X)| | Y = y] < \infty$ a.s. for $y \in \mathcal{Y}$.

Let $\nu : y \mapsto \mathbb{E}[m(X)|Y = y]$. We can clearly rewrite $g$ as the sum of $\nu$ and a centered GP on $\mathcal{Y}$

$$g(y) = \nu(y) + \int_{\mathcal{X}} (f(x) - m(x))\, d\mathbb{P}_{X|Y=y}(x), \qquad \forall y \in \mathcal{Y} \tag{27}$$

which is well-defined almost surely.

It hence comes $\mathbb{E}[g(y)] = \mathbb{E}[\nu(y)] + 0 = \nu(y)$. Plus since $\nu(y)$ is a constant shift, the covariance is not affected and has the same expression than for the centered GP. Since this holds for any $y \in \mathcal{Y}$, we conclude that $g \sim \mathcal{GP}(\nu, q)$ a.s.

**Show that $q(y, y') = \langle \mu_{X|Y=y}, \mu_{X|Y=y'} \rangle_k$** First, we know by Proposition A.1 that $\mathbb{E}[\|k_X\|_k | Y = y] < \infty$ $\mathbb{P}_Y$-a.e. .By triangular inequality, we obtain $\|\mu_{X|Y=y}\|_k = \|\mathbb{E}[k_X | Y = y]\|_k \leq \mathbb{E}[\|k_X\|_k | Y = y] < \infty$ $\mathbb{P}_Y$-a.e. and hence $\mu_{X|Y=y}$ is well-defined up to a set of measure zero with respect to $\mathbb{P}_Y$.

With notations from Proposition 3.2, we can proceed for any $y, y' \in \mathcal{Y}$ as

$$q(y, y') = \mathbb{E}[k(X, X')|Y = y', Y' = y'] \tag{28}$$

$$= \int_{\mathcal{X}} \int_{\mathcal{X}} k(x, x') \, d\mathbb{P}_{X|Y=y}(x) \, d\mathbb{P}_{X|Y=y'}(x') \tag{29}$$

$$= \int_{\mathcal{X}} \int_{\mathcal{X}} \langle k_x, k_{x'} \rangle_k \, d\mathbb{P}_{X|Y=y}(x) \, d\mathbb{P}_{X|Y=y'}(x') \tag{30}$$

$$= \left\langle \int_{\mathcal{X}} k_x \, d\mathbb{P}_{X|Y=y}(x), \int_{\mathcal{X}} k_{x'} \, d\mathbb{P}_{X|Y=y'}(x') \right\rangle_k \quad \text{a.s.} \tag{31}$$

$$= \langle \mu_{X|Y=y}, \mu_{X|Y=y'} \rangle_k \quad \text{a.s.} \tag{32}$$

$$\square$$

**Proposition 3.3.** *Given aggregate observations $\tilde{\mathbf{z}}$ with homoscedastic noise $\sigma^2$, the deconditional posterior of $f$ is defined as the Gaussian process $f|\tilde{\mathbf{z}} \sim \mathcal{GP}(m_{\mathrm{d}}, k_{\mathrm{d}})$ where*

$$m_{\mathrm{d}}(x) = m(x) + k_x^\top C_{X|Y} \boldsymbol{\Psi}_{\tilde{\mathbf{y}}} (\mathbf{Q}_{\tilde{\mathbf{y}}\tilde{\mathbf{y}}} + \sigma^2 \mathbf{I}_M)^{-1} (\tilde{\mathbf{z}} - \nu(\tilde{\mathbf{y}})), \tag{33}$$

$$k_{\mathrm{d}}(x, x') = k(x, x') - k_x^\top C_{X|Y} \boldsymbol{\Psi}_{\tilde{\mathbf{y}}} (\mathbf{Q}_{\tilde{\mathbf{y}}\tilde{\mathbf{y}}} + \sigma^2 \mathbf{I}_M)^{-1} \boldsymbol{\Psi}_{\tilde{\mathbf{y}}}^\top C_{X|Y}^\top k_{x'}. \tag{34}$$

*Proof of Proposition 3.3.* Recall that

$$\begin{bmatrix} f(\mathbf{x}) \\ \tilde{\mathbf{z}} \end{bmatrix} | \mathbf{y}, \tilde{\mathbf{y}} \sim \mathcal{N} \left( \begin{bmatrix} m(\mathbf{x}) \\ \nu(\tilde{\mathbf{y}}) \end{bmatrix}, \begin{bmatrix} \mathbf{K}_{\mathbf{xx}} & \boldsymbol{\Upsilon} \\ \boldsymbol{\Upsilon}^\top & \mathbf{Q}_{\tilde{\mathbf{y}}\tilde{\mathbf{y}}} + \sigma^2 \mathbf{I}_M \end{bmatrix} \right). \tag{35}$$

where $\boldsymbol{\Upsilon} = \mathrm{Cov}(f(\mathbf{x}), \tilde{\mathbf{z}}) = \boldsymbol{\Phi}_{\mathbf{x}}^\top C_{X|Y} \boldsymbol{\Psi}_{\tilde{\mathbf{y}}}$.

Applying Gaussian conditioning, we obtain that

$$f(\mathbf{x}) | \tilde{\mathbf{z}}, \mathbf{y}, \tilde{\mathbf{y}} \sim \mathcal{N}(m(\mathbf{x}) + \boldsymbol{\Upsilon} (\mathbf{Q}_{\tilde{\mathbf{y}}\tilde{\mathbf{y}}} + \sigma^2 \mathbf{I}_M)^{-1} (\tilde{\mathbf{z}} - \nu(\tilde{\mathbf{y}})), \tag{36}$$

$$\mathbf{K}_{\mathbf{xx}} - \boldsymbol{\Upsilon} (\mathbf{Q}_{\tilde{\mathbf{y}}\tilde{\mathbf{y}}} + \sigma^2 \mathbf{I}_M)^{-1} \boldsymbol{\Upsilon}^\top) \tag{37}$$

Since the latter holds for any input $\mathbf{x} \in \mathcal{X}^N$, by Kolmogorov extension theorem this implies that $f$ conditioned on the data $\tilde{\mathbf{z}}, \tilde{\mathbf{y}}$ is a draw from a GP. We denote it $f|\tilde{\mathbf{z}} \sim \mathcal{GP}(m_{\mathrm{d}}, k_{\mathrm{d}})$ and it is specified by

$$m_{\mathrm{d}}(x) = m(x) + k_x^\top C_{X|Y} \boldsymbol{\Psi}_{\tilde{\mathbf{y}}} (\mathbf{Q}_{\tilde{\mathbf{y}}\tilde{\mathbf{y}}} + \sigma^2 \mathbf{I}_M)^{-1} (\tilde{\mathbf{z}} - \nu(\tilde{\mathbf{y}})), \tag{38}$$

$$k_{\mathrm{d}}(x, x') = k(x, x') - k_x^\top C_{X|Y} \boldsymbol{\Psi}_{\tilde{\mathbf{y}}} (\mathbf{Q}_{\tilde{\mathbf{y}}\tilde{\mathbf{y}}} + \sigma^2 \mathbf{I}_M)^{-1} \boldsymbol{\Psi}_{\tilde{\mathbf{y}}}^\top C_{X|Y}^\top k_{x'}. \tag{39}$$

Note that we abuse notation

$$"k_x^\top C_{X|Y} \boldsymbol{\Psi}_{\tilde{\mathbf{y}}}" = \begin{bmatrix} \langle k_x, C_{X|Y} \ell_{\tilde{y}_1} \rangle_k & \cdots & \langle k_x, C_{X|Y} \ell_{\tilde{y}_M} \rangle_k \end{bmatrix} \tag{40}$$

$$= \begin{bmatrix} \langle k_x, \mu_{X|Y=\tilde{y}_1} \rangle_k & \cdots & \langle k_x, \mu_{X|Y=\tilde{y}_M} \rangle_k \end{bmatrix} \tag{41}$$

$$= \begin{bmatrix} \mathrm{Cov}(f(x), g(\tilde{y}_1)) & \cdots & \mathrm{Cov}(f(x), g(\tilde{y}_M)) \end{bmatrix}. \tag{42}$$

$$\square$$

## A.2 Proofs of Section 4

**Proposition 4.1** (Empirical DMO as vector-valued regressor)**.** *The minimiser of the empirical reconstruction risk is the empirical DMO, i.e. $\hat{D}_{X|Y} = \arg\min_{D \in \mathcal{H}_\Gamma} \hat{\mathcal{E}}_{\mathrm{d}}(D)$*

*Proof of Proposition 4.1.* Let $D \in \mathcal{H}_\Gamma$, we recall the form of the regularised empirical objective

$$\hat{\mathcal{E}}_{\mathrm{d}}(D) = \frac{1}{M} \sum_{j=1}^{M} \|\ell_{\tilde{y}_j} - D\hat{C}_{X|Y} \ell_{\tilde{y}_j}\|_\ell^2 + \epsilon \|D\|_\Gamma^2 \tag{43}$$

By [46, Theorem 4.1], if $\hat{D} \in \arg\min\limits_{D \in \mathcal{H}_\Gamma} \hat{\mathcal{E}}_d(D)$, then it is unique and has form

$$\hat{D} = \sum_{j=1}^{M} \Gamma_{\hat{C}_{X|Y}\ell_{\tilde{y}_j}} c_j \tag{44}$$

where $\Gamma_{\hat{C}_{X|Y}\ell_{\tilde{y}_j}} : \mathcal{H}_\ell \to \mathcal{H}_\Gamma$ is the vector-valued kernel $\Gamma$'s feature map indexed by $\hat{C}_{X|Y}\ell_{\tilde{y}_j}$, such that for any $h \in \mathcal{H}_\Gamma$ and $g \in \mathcal{H}_\ell$, we have $\langle h, \Gamma_{\hat{C}_{X|Y}\ell_{\tilde{y}_j}} g \rangle_\Gamma = \langle h(\hat{C}_{X|Y}\ell_{\tilde{y}_j}), g \rangle_\ell$. (see [35] for a detailed review of vector-valued RKHS). Furthermore, coefficients $c_1, \ldots, c_M \in \mathcal{H}_\ell$ are the unique solutions to

$$\sum_{i=1}^{M} \left( \Gamma(\hat{C}_{X|Y}\ell_{\tilde{y}_i}, \hat{C}_{X|Y}\ell_{\tilde{y}_j}) + M\epsilon\delta_{ij} \right) c_i = \ell_{\tilde{y}_j} \tag{45}$$

Since

$$\Gamma(\hat{C}_{X|Y}\ell_{\tilde{y}_i}, \hat{C}_{X|Y}\ell_{\tilde{y}_j}) = \langle \hat{C}_{X|Y}\ell_{\tilde{y}_i}, \hat{C}_{X|Y}\ell_{\tilde{y}_j} \rangle_k \operatorname{Id}_{\mathcal{H}_\ell} = \hat{q}(\tilde{y}_i, \tilde{y}_j) \operatorname{Id}_{\mathcal{H}_\ell} \tag{46}$$

where $\operatorname{Id}_{\mathcal{H}_\ell}$ denotes the identity operator on $\mathcal{H}_\ell$. The above simplifies as

$$\sum_{i=1}^{M} \left( \hat{q}(\tilde{y}_i, \tilde{y}_j) + M\epsilon\delta_{ij} \right) c_i = \ell_{\tilde{y}_j} \quad \forall 1 \leq j \leq M \tag{47}$$

$$\Leftrightarrow \left( \hat{\mathbf{Q}}_{\tilde{\mathbf{y}}\tilde{\mathbf{y}}} + M\epsilon\mathbf{I}_M \right) \mathbf{c}^\top = \mathbf{\Psi}_{\tilde{\mathbf{y}}}^\top \tag{48}$$

$$\Leftrightarrow \mathbf{c} = \mathbf{\Psi}_{\tilde{\mathbf{y}}} \left( \hat{\mathbf{Q}}_{\tilde{\mathbf{y}}\tilde{\mathbf{y}}} + M\epsilon\mathbf{I}_M \right)^{-1} \tag{49}$$

where $\mathbf{c} = [c_1 \quad \ldots \quad c_M]$.

Since for any $f \in \mathcal{H}_k$ and $g \in \mathcal{H}_\ell$, our choice of kernel gives $\Gamma_f g = g \otimes f$, plugging (47) into (44) we obtain

$$\hat{D} = \sum_{j=1}^{M} \Gamma_{\hat{C}_{X|Y}\ell_{\tilde{y}_j}} c_j \tag{50}$$

$$= \sum_{j=1}^{M} c_j \otimes \hat{C}_{X|Y}\ell_{\tilde{y}_j} \tag{51}$$

$$= \mathbf{c} \left[ \hat{C}_{X|Y}\mathbf{\Psi}_{\tilde{\mathbf{y}}} \right]^\top \tag{52}$$

$$= \left[ \mathbf{\Psi}_{\tilde{\mathbf{y}}} \left( \hat{\mathbf{Q}}_{\tilde{\mathbf{y}}\tilde{\mathbf{y}}} + M\epsilon\mathbf{I}_M \right)^{-1} \right] \left[ \hat{C}_{X|Y}\mathbf{\Psi}_{\tilde{\mathbf{y}}} \right]^\top \tag{53}$$

$$= \mathbf{\Psi}_{\tilde{\mathbf{y}}} \left( \hat{\mathbf{Q}}_{\tilde{\mathbf{y}}\tilde{\mathbf{y}}} + M\epsilon\mathbf{I}_M \right)^{-1} \mathbf{\Psi}_{\tilde{\mathbf{y}}}^\top \hat{C}_{X|Y}^\top \tag{54}$$

$$= \mathbf{\Psi}_{\tilde{\mathbf{y}}} \left( \hat{\mathbf{Q}}_{\tilde{\mathbf{y}}\tilde{\mathbf{y}}} + M\epsilon\mathbf{I}_M \right)^{-1} \mathbf{\Psi}_{\tilde{\mathbf{y}}}^\top \mathbf{\Psi}_{\mathbf{y}} \left( \mathbf{L}_{\mathbf{yy}} + N\lambda\mathbf{I}_N \right)^{-1} \mathbf{\Phi}_{\mathbf{x}} \tag{55}$$

$$= \mathbf{\Psi}_{\tilde{\mathbf{y}}} \left( \hat{\mathbf{Q}}_{\tilde{\mathbf{y}}\tilde{\mathbf{y}}} + M\epsilon\mathbf{I}_M \right)^{-1} \mathbf{A}\mathbf{\Phi}_{\mathbf{x}} \tag{56}$$

$$= \hat{D}_{X|Y} \tag{57}$$

which concludes the proof. $\qquad\square$

**Theorem 4.2** (Empirical DMO Convergence Rate). *Denote $D_{\mathbb{P}_Y} = \arg\min_{D \in \mathcal{H}_\Gamma} \mathcal{E}_d(D)$. Assume assumptions stated in Appendix D are satisfied. In particular, let $(b, c)$ and $(0, c')$ be the parameters of the restricted class of distribution for $\mathbb{P}_Y$ and $\mathbb{P}_{XY}$ respectively and let $\iota \in \,]0, 1]$ be the Hölder continuity exponent in $\mathcal{H}_\Gamma$. Then, if we choose $\lambda = N^{-\frac{1}{c'+1}}$, $N = M^{\frac{a(c'+1)}{\iota(c'-1)}}$ where $a > 0$, we have the following result,*

- *If $a \leq \frac{b(c+1)}{bc+1}$, then $\mathcal{E}_d(\hat{D}_{X|Y}) - \mathcal{E}_d(D_{\mathbb{P}_Y}) = \mathcal{O}(M^{\frac{-ac}{c+1}})$ with $\epsilon = M^{\frac{-a}{c+1}}$*

- *If $a \geq \frac{b(c+1)}{bc+1}$, then $\mathcal{E}_{\mathrm{d}}(\hat{D}_{X|Y}) - \mathcal{E}_{\mathrm{d}}(D_{\mathbb{P}_Y}) = \mathcal{O}(M^{\frac{-bc}{bc+1}})$ with $\epsilon = M^{\frac{-b}{bc+1}}$*

*Proof of Theorem 4.2.* In Appendix D, we present Theorem D.4 which is a detailed version of this result with all assumptions explicitly stated. The proof of Theorem D.4 constitutes the proof of this result. ∎

# B  Variational formulation of the deconditional posterior

Inference computational complexity is $\mathcal{O}(M^3)$ for the posterior mean and $\mathcal{O}(N^3 + M^3)$ for the posterior covariance. To scale to large datasets, we introduce in the following a variational formulation as a scalable approximation to the deconditional posterior $f(\mathbf{x})|\tilde{\mathbf{z}}$. Without loss of generality, we assume in the following that $f$ is centered, i.e. $m = 0$.

## B.1  Variational formulation

We consider a set of $d$ inducing locations $\mathbf{w} = [w_1 \quad \ldots \quad w_d]^\top \in \mathcal{X}^d$ and define inducing points as the gaussian vector $\mathbf{u} := f(\mathbf{w}) \sim \mathcal{N}(0, \mathbf{K_{ww}})$, where $\mathbf{K_{ww}} := k(\mathbf{w}, \mathbf{w})$. We set $d$-dimensional variational distribution $q(\mathbf{u}) = \mathcal{N}(\mathbf{u}|\boldsymbol{\eta}, \boldsymbol{\Sigma})$ over inducing points and define $q(\mathbf{f}) := \int p(\mathbf{f}|\mathbf{u})q(\mathbf{u})\,\mathrm{d}\mathbf{u}$ as an approximation of the deconditional posterior $p(\mathbf{f}|\mathbf{z})$. The estimation of the deconditional posterior can thus be approximated by optimising the variational distribution parameters $\boldsymbol{\eta}, \boldsymbol{\Sigma}$ to maximise the *evidence lower bound* (ELBO) objective given by

$$\mathrm{ELBO}(q) = \mathbb{E}_{q(\mathbf{f})}[\log p(\tilde{\mathbf{z}}|\mathbf{f})] + \mathrm{KL}(q(\mathbf{u})\|p(\mathbf{u})). \tag{58}$$

As both $q$ and $p$ are Gaussians, the Kullback-Leibler divergence admits closed-form. The expected log likelihood term decomposes as

$$\mathbb{E}_{q(\mathbf{f})}[\log p(\mathbf{z}|\mathbf{f})] = -\frac{M}{2}\log(2\pi\sigma^2) + \frac{1}{2\sigma^2}\left(\mathrm{tr}\left(\mathbf{A}^\top\bar{\boldsymbol{\Sigma}}\mathbf{A}\right) + \left\|\tilde{\mathbf{z}} - A^\top\bar{\boldsymbol{\eta}}\right\|_2^2\right) \tag{59}$$

where $\bar{\boldsymbol{\eta}}$ and $\bar{\boldsymbol{\Sigma}}$ are the parameters of the posterior variational distribution $q(\mathbf{f}) = \mathcal{N}(\mathbf{f}|\bar{\boldsymbol{\eta}}, \bar{\boldsymbol{\Sigma}})$ given by

$$\bar{\boldsymbol{\eta}} = \mathbf{K_{xw}}\mathbf{K_{ww}^{-1}}\boldsymbol{\eta} \qquad \bar{\boldsymbol{\Sigma}} = \mathbf{K_{xx}} - \mathbf{K_{xw}}\left[\mathbf{K_{ww}^{-1}} - \mathbf{K_{ww}^{-1}}\boldsymbol{\Sigma}\mathbf{K_{ww}^{-1}}\right]\mathbf{K_{wx}} \tag{60}$$

Given this objective, we can optimise this lower bound with respect to variational parameters $\boldsymbol{\eta}, \boldsymbol{\Sigma}$, noise $\sigma^2$ and parameters of kernels $k$ and $\ell$, with an option to parametrize these kernels using feature maps given by deep neural network [47], using a stochastic gradient approach for example. We might also want to learn the inducing locations $\mathbf{w}$.

## B.2  Details on evidence lower bound derivation

For completeness, we provide here the derivation of the evidence lower bound objective. Let us remind its expression as stated in (58)

$$\mathrm{ELBO}(q) = \mathbb{E}_{q(\mathbf{f})}[\log p(\tilde{\mathbf{z}}|\mathbf{f})] - \mathrm{KL}(q(\mathbf{u})\|p(\mathbf{u})) \tag{61}$$

The second term here is the Kullback-Leibler divergence of two gaussian densities which has a known and tractable closed-form expression.

$$\mathrm{KL}(q(\mathbf{u})\|p(\mathbf{u})) = \frac{1}{2}\left[\mathrm{tr}\left(\mathbf{K_{ww}^{-1}}\boldsymbol{\Sigma}\right) + \boldsymbol{\eta}^\top\mathbf{K_{ww}^{-1}}\boldsymbol{\eta} - d + \log\frac{\det\mathbf{K_{ww}}}{\det\boldsymbol{\Sigma}}\right] \tag{62}$$

The first term is the expected log likelihood and needs to be derived. Using properties of integrals of gaussian densities, we can start by showing that $q(\mathbf{f})$ also corresponds to a gaussian density which comes

$$q(\mathbf{f}) = \int p(\mathbf{f}|\mathbf{u})q(\mathbf{u})\,\mathrm{d}\mathbf{u} \tag{63}$$

$$= \int \mathcal{N}(\mathbf{f}|\mathbf{K_{xw}}\mathbf{K_{ww}^{-1}}\mathbf{u}, \mathbf{K_{xx}} - \mathbf{K_{xw}}\mathbf{K_{ww}^{-1}}\mathbf{K_{wx}}) \times \mathcal{N}(\mathbf{u}|\boldsymbol{\eta}, \boldsymbol{\Sigma})\,\mathrm{d}\mathbf{u} \tag{64}$$

$$= \mathcal{N}(\mathbf{f}|\bar{\boldsymbol{\eta}}, \bar{\boldsymbol{\Sigma}}) \tag{65}$$

where

$$\bar{\boldsymbol{\eta}} = \mathbf{K_{xw}}\mathbf{K_{ww}^{-1}}\boldsymbol{\eta} \tag{66}$$

$$\bar{\boldsymbol{\Sigma}} = \mathbf{K_{xx}} - \mathbf{K_{xw}}\left[\mathbf{K_{ww}^{-1}} - \mathbf{K_{ww}^{-1}}\boldsymbol{\Sigma}\mathbf{K_{ww}^{-1}}\right]\mathbf{K_{wx}} \tag{67}$$

Let's try now to obtain a closed-form expression of $\mathbb{E}_{q(\mathbf{f})}[\log p(\tilde{\mathbf{z}}|\mathbf{f})]$ on which we will be able to perform a gradient-based optimization routine. Using Gaussian conditioning on (5), we obtain

$$p(\tilde{\mathbf{z}}|\mathbf{f}) = \mathcal{N}(\tilde{\mathbf{z}}|\mathbf{\Upsilon}^\top \mathbf{K}_{\mathbf{xx}}^{-1}\mathbf{f} , \ \mathbf{Q}_{\tilde{\mathbf{y}}\tilde{\mathbf{y}}} + \sigma^2 \mathbf{I}_M - \mathbf{\Upsilon}^\top \mathbf{K}_{\mathbf{xx}}^{-1}\mathbf{\Upsilon}) \tag{68}$$

We notice that $\mathbf{\Upsilon}^\top \mathbf{K}_{\mathbf{xx}}^{-1} = \ell(\tilde{\mathbf{y}}, \mathbf{y})(\mathbf{L}_{\mathbf{yy}} + \lambda N \mathbf{I}_N)^{-1}\mathbf{K}_{\mathbf{xx}}\mathbf{K}_{\mathbf{xx}}^{-1} = \ell(\tilde{\mathbf{y}}, \mathbf{y})(\mathbf{L}_{\mathbf{yy}} + \lambda N \mathbf{I}_N)^{-1} = \mathbf{A}$.

Hence we also have $\mathbf{\Upsilon}^\top \mathbf{K}_{\mathbf{xx}}^{-1}\mathbf{\Upsilon} = \mathbf{A}^\top \mathbf{K}_{\mathbf{xx}}\mathbf{A} = \mathbf{Q}_{\tilde{\mathbf{y}}\tilde{\mathbf{y}}}$.

We can thus simplify (68) as

$$p(\tilde{\mathbf{z}}|\mathbf{f}) = \mathcal{N}(\tilde{\mathbf{z}}|\mathbf{A}^\top \mathbf{f}, \sigma^2 \mathbf{I}_n) \tag{69}$$

Then,

$$\log p(\tilde{\mathbf{z}}|\mathbf{f}) = -\frac{M}{2}\log(2\pi\sigma^2) - \frac{1}{2\sigma^2}\left\|\tilde{\mathbf{z}} - \mathbf{A}^\top \mathbf{f}\right\|_2^2 \tag{70}$$

$$\Rightarrow \mathbb{E}_{q(\mathbf{f})}[\log p(\tilde{\mathbf{z}}|\mathbf{f})] = -\frac{M}{2}\log(2\pi\sigma^2) - \frac{1}{2\sigma^2}\mathbb{E}_{q(\mathbf{f})}\left[\left\|\tilde{\mathbf{z}} - \mathbf{A}^\top \mathbf{f}\right\|_2^2\right] \tag{71}$$

Using the trace trick to express the expectation with respect to the posterior variational parameters $\bar{\boldsymbol{\eta}}, \bar{\boldsymbol{\Sigma}}$, we have

$$\mathbb{E}_{q(\mathbf{f})}\left[\left\|\tilde{\mathbf{z}} - \mathbf{A}^\top \mathbf{f}\right\|_2^2\right] = \mathbb{E}_{q(\mathbf{f})}\left[\mathrm{tr}\left(\left(\tilde{\mathbf{z}} - \mathbf{A}^\top \mathbf{f}\right)^\top \left(\tilde{\mathbf{z}} - \mathbf{A}^\top \mathbf{f}\right)\right)\right] \tag{72}$$

$$= \mathbb{E}_{q(\mathbf{f})}\left[\mathrm{tr}\left(\left(\tilde{\mathbf{z}} - \mathbf{A}^\top \mathbf{f}\right)\left(\tilde{\mathbf{z}} - \mathbf{A}^\top \mathbf{f}\right)^\top\right)\right] \tag{73}$$

$$= \mathrm{tr}\left(\mathbb{E}_{q(\mathbf{f})}\left[\left(\tilde{\mathbf{z}} - \mathbf{A}^\top \mathbf{f}\right)\left(\tilde{\mathbf{z}} - \mathbf{A}^\top \mathbf{f}\right)^\top\right]\right) \tag{74}$$

$$\tag{75}$$

And

$$\mathbb{E}_{q(\mathbf{f})}\left[\left(\tilde{\mathbf{z}} - \mathbf{A}^\top \mathbf{f}\right)\left(\tilde{\mathbf{z}} - \mathbf{A}^\top \mathbf{f}\right)^\top\right] = \mathrm{Cov}(\tilde{\mathbf{z}} - \mathbf{A}^\top \mathbf{f}) + \mathbb{E}_{q(\mathbf{f})}\left[\tilde{\mathbf{z}} - \mathbf{A}^\top \mathbf{f}\right]\mathbb{E}_{q(\mathbf{f})}\left[\tilde{\mathbf{z}} - \mathbf{A}^\top \mathbf{f}\right]^\top \tag{76}$$

$$= \mathbf{A}^\top \bar{\boldsymbol{\Sigma}}\mathbf{A} + \left(\tilde{\mathbf{z}} - \mathbf{A}^\top \bar{\boldsymbol{\eta}}\right)\left(\tilde{\mathbf{z}} - \mathbf{A}^\top \bar{\boldsymbol{\eta}}\right)^\top \tag{77}$$

Hence, it comes that

$$\mathbb{E}_{q(\mathbf{f})}\left[\left\|\tilde{\mathbf{z}} - \mathbf{A}^\top \mathbf{f}\right\|_2^2\right] = \mathrm{tr}\left(\mathbf{A}^\top \bar{\boldsymbol{\Sigma}}\mathbf{A}\right) + \mathrm{tr}\left(\left(\tilde{\mathbf{z}} - \mathbf{A}^\top \bar{\boldsymbol{\eta}}\right)\left(\tilde{\mathbf{z}} - \mathbf{A}^\top \bar{\boldsymbol{\eta}}\right)^\top\right) \tag{78}$$

$$= \mathrm{tr}\left(\mathbf{A}^\top \bar{\boldsymbol{\Sigma}}\mathbf{A}\right) + \left\|\tilde{\mathbf{z}} - \mathbf{A}^\top \bar{\boldsymbol{\eta}}\right\|_2^2 \tag{79}$$

which can be efficiently computed as it only requires diagonal terms.

Wrapping up, we obtain that

$$\mathrm{ELBO}(q) = -\frac{M}{2}\log(2\pi\sigma^2) - \frac{1}{2\sigma^2}\left(\mathrm{tr}\left(\mathbf{A}^\top \bar{\boldsymbol{\Sigma}}\mathbf{A}\right) + \left\|\tilde{\mathbf{z}} - \mathbf{A}^\top \bar{\boldsymbol{\eta}}\right\|_2^2\right) - \mathrm{KL}(q(\mathbf{u})\|p(\mathbf{u})) \tag{80}$$

# C  Details on Conditional Mean Shrinkage Operator

## C.1  Deconditional posterior with Conditional Mean Shrinkage Operator

We recall from Proposition 3.3 that the deconditional posterior is a GP specifed by mean and covariance functions

$$m_{\mathrm{d}}(x) = m(x) + k_x^\top C_{X|Y} \mathbf{\Psi}_{\tilde{\mathbf{y}}} (\mathbf{Q}_{\tilde{\mathbf{y}}\tilde{\mathbf{y}}} + \sigma^2 \mathbf{I}_M)^{-1}(\tilde{\mathbf{z}} - \nu(\tilde{\mathbf{y}})), \tag{81}$$

$$k_{\mathrm{d}}(x,x') = k(x,x') - k_x^\top C_{X|Y} \mathbf{\Psi}_{\tilde{\mathbf{y}}} (\mathbf{Q}_{\tilde{\mathbf{y}}\tilde{\mathbf{y}}} + \sigma^2 \mathbf{I}_M)^{-1} \mathbf{\Psi}_{\tilde{\mathbf{y}}}^\top C_{X|Y}^\top k_{x'} \tag{82}$$

for any $x, x' \in \mathcal{X}$, where we abuse notation for the cross-covariance term

$$"k_x^\top C_{X|Y} \mathbf{\Psi}_{\tilde{\mathbf{y}}}" = \begin{bmatrix} \langle k_x, C_{X|Y} \ell_{\tilde{y}_1} \rangle_k & \cdots & \langle k_x, C_{X|Y} \ell_{\tilde{y}_M} \rangle_k \end{bmatrix}. \tag{83}$$

The CMO appears in the cross-covariance term $k_x^\top C_{X|Y} \mathbf{\Psi}_{\tilde{\mathbf{y}}}$ and in the CMP covariance matrix $\mathbf{Q}_{\tilde{\mathbf{y}}\tilde{\mathbf{y}}} = \mathbf{\Psi}_{\tilde{\mathbf{y}}}^\top C_{X|Y}^\top C_{X|Y} \mathbf{\Psi}_{\tilde{\mathbf{y}}}$. To derive empirical versions using the Conditional Mean Shrinkage Operator we replace it by ${}^S \hat{C}_{X|Y} = \hat{\mathbf{M}}_{\mathbf{y}} (\mathbf{L}_{\mathbf{yy}} + \lambda N \mathbf{I}_N)^{-1} \mathbf{\Psi}_{\mathbf{y}}^\top$.

The empirical cross-covariance operator with shrinkage CMO estimate is given by

$$k_x^\top {}^S \hat{C}_{X|Y} \mathbf{\Psi}_{\tilde{\mathbf{y}}} = k_x^\top \hat{\mathbf{M}}_{\mathbf{y}} (\mathbf{L}_{\mathbf{yy}} + \lambda N \mathbf{I}_N)^{-1} \mathbf{\Psi}_{\mathbf{y}}^\top \mathbf{\Psi}_{\tilde{\mathbf{y}}} \tag{84}$$

$$= k_x^\top \hat{\mathbf{M}}_{\mathbf{y}} (\mathbf{L}_{\mathbf{yy}} + \lambda N \mathbf{I}_N)^{-1} \mathbf{L}_{\mathbf{y}\tilde{\mathbf{y}}} \tag{85}$$

$$= k_x^\top \hat{\mathbf{M}}_{\mathbf{y}} \mathbf{A} \tag{86}$$

where we abuse notation

$$"k_x^\top \hat{\mathbf{M}}_{\mathbf{y}}" := \begin{bmatrix} \langle k_x, \hat{\mu}_{X|Y=y_1} \rangle_k & \cdots & \langle k_x, \hat{\mu}_{X|Y=y_N} \rangle_k \end{bmatrix} \tag{87}$$

$$= \begin{bmatrix} \frac{1}{n_1} \sum_{i=1}^{n_1} k(x_1^{(i)}, x) & \cdots & \frac{1}{n_N} \sum_{i=1}^{n_N} k(x_N^{(i)}, x) \end{bmatrix}. \tag{88}$$

The empirical shrinkage CMP covariance matrix is given by

$${}^S \hat{\mathbf{Q}}_{\tilde{\mathbf{y}}\tilde{\mathbf{y}}} := \mathbf{\Psi}_{\tilde{\mathbf{y}}}^\top {}^S \hat{C}_{X|Y}^\top {}^S \hat{C}_{X|Y} \mathbf{\Psi}_{\tilde{\mathbf{y}}} \tag{89}$$

$$= \mathbf{\Psi}_{\tilde{\mathbf{y}}}^\top \mathbf{\Psi}_{\mathbf{y}} (\mathbf{L}_{\mathbf{yy}} + \lambda N \mathbf{I}_N)^{-1} \hat{\mathbf{M}}_{\mathbf{y}}^\top \hat{\mathbf{M}}_{\mathbf{y}} (\mathbf{L}_{\mathbf{yy}} + \lambda N \mathbf{I}_N)^{-1} \mathbf{\Psi}_{\mathbf{y}}^\top \mathbf{\Psi}_{\tilde{\mathbf{y}}} \tag{90}$$

$$= \mathbf{A}^\top \hat{\mathbf{M}}_{\mathbf{y}}^\top \hat{\mathbf{M}}_{\mathbf{y}} \mathbf{A} \tag{91}$$

where with similar notation abuse

$$"\hat{\mathbf{M}}_{\mathbf{y}}^\top \hat{\mathbf{M}}_{\mathbf{y}}" = \begin{bmatrix} \langle \hat{\mu}_{X|Y=y_i}, \hat{\mu}_{X|Y=y_j} \rangle_k \end{bmatrix}_{1 \leq i,j \leq N} = \begin{bmatrix} \frac{1}{n_i n_j} \sum_{l=1}^{n_i} \sum_{r=1}^{n_j} k(x_i^{(l)}, x_j^{(r)}) \end{bmatrix}_{1 \leq i,j \leq N} \tag{92}$$

Substituting the latters into (81) and (82), we obtain empirical estimates of the deconditional posterior with shrinkage CMO estimator defined as

$${}^S \hat{m}_{\mathrm{d}}(x) := m(x) + k_x^\top \hat{\mathbf{M}}_{\mathbf{y}} \mathbf{A} (\mathbf{A}^\top \hat{\mathbf{M}}_{\mathbf{y}}^\top \hat{\mathbf{M}}_{\mathbf{y}} \mathbf{A} + \sigma^2 \mathbf{I}_M)^{-1}(\tilde{\mathbf{z}} - \hat{\mu}(\tilde{\mathbf{y}})), \tag{93}$$

$${}^S \hat{k}_{\mathrm{d}}(x,x') := k(x,x') - k_x^\top \hat{\mathbf{M}}_{\mathbf{y}} \mathbf{A} (\mathbf{A}^\top \hat{\mathbf{M}}_{\mathbf{y}}^\top \hat{\mathbf{M}}_{\mathbf{y}} \mathbf{A} + \sigma^2 \mathbf{I}_M)^{-1} \mathbf{A}^\top \hat{\mathbf{M}}_{\mathbf{y}}^\top k_{x'} \tag{94}$$

for any $x, x' \in \mathcal{X}$.

Note that as the number of bags increases, it is possible to derive a variational formulation similar to the one proposed in Section B that leverages the shrinkage estimator to further speed up the overall computation.

## C.2 Ablation Study

In this section we will present an ablation study on the shrinkage CMO estimator. The key is to illustrate that the Shrinkage CMO performs on par with the standard CMO estimator but is much faster to compute.

In the following, we will sample bag data of the form ${}^b\mathcal{D} = \{{}^b\boldsymbol{x}_j, y_j\}_{j=1}^N$ and ${}^b\boldsymbol{x}_j = \{x_j^{(i)}\}_{i=1}^n$, i.e there are $N$ bags with $n$ elements inside each. We first sample $N$ bag labels $y_j \sim \mathcal{N}(0, 2)$ and for each bag $y_j$, we sample $n$ observations $x_j^{(i)}|y_j \sim \mathcal{N}(y_j \sin(y_j), 0.5^2)$.

Recall in standard CME one would need to repeat the number of bag labels to match the cardinality of $x_j^{(i)}$, i.e estimating CME using data $\{x_j^{(i)}, y_j\}_{j=1,i=1}^{N,n}$.

Denote $\hat{C}_{X|Y}$ as the standard CMO estimator and ${}^S\hat{C}_{X|Y}$ as the shrinkage CMO estimator. We will compare the RMSE between the two estimator when tested on a grid of test points $\{x_i^*, y_i^*\}_{i=1}^{N^*}$, i.e comparing the RMSE of the values between $\hat{\mu}_{X|Y=y_i^*}(x_i^*) := \langle \hat{C}_{X|Y}\ell_{y_i^*}, k_{x_i^*} \rangle_k$ and ${}^S\hat{\mu}_{X|Y=y_i^*}(x_i^*) := \langle {}^S\hat{C}_{X|Y}\ell_{y_i^*}, k_{x_i^*} \rangle_k$ for each $i$. We also report the time in seconds needed to compute the estimator. The following results are ran on a CPU. Kernel hyperparameters are chosen using the median heuristic. The regularisation for both estimator is set to $0.1$.

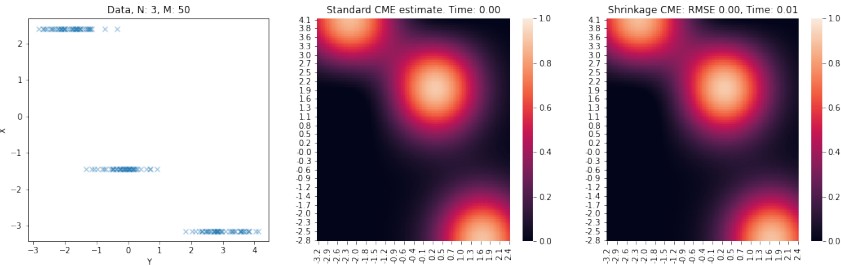

Figure 4: 3 bags with 50 samples each. (left) Data, (middle) $\hat{\mu}_{X|Y=y_i^*}(x_i^*)$ Standard CME. (right) ${}^S\hat{\mu}_{X|Y=y_i^*}(x_i^*)$ Shrinkage CME. We see both algorithms require very little time to train, ($\sim 0.01$second) with a negligible difference in values as shown by the RMSE.

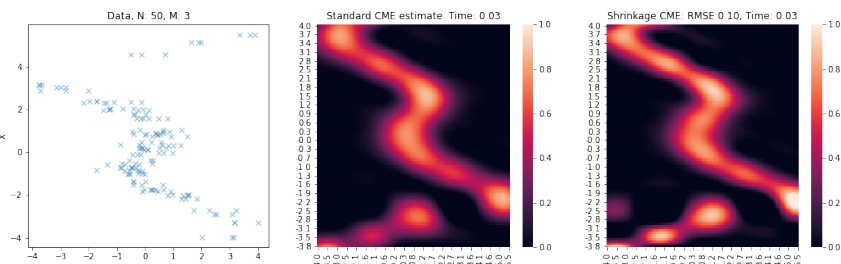

Figure 5: 50 bags with 3 samples each. (left) Data, (middle) $\hat{\mu}_{X|Y=y_i^*}(x_i^*)$ Standard CME. (right) ${}^S\hat{\mu}_{X|Y=y_i^*}(x_i^*)$ Shrinkage CME. Again, we see both algorithms require very little time to train, ($\sim 0.03$ second). However, there is an increase in RMSE for the shrinkage estimator because there are much less samples for each bag, thus the empirical CME estimate $\hat{\mu}_{X|Y=y_j}$ might not be accurate. Nonetheless, it is still a small difference.

Figures 4 and 5 show how shrinkage CMO performed compared to the standard CMO in a small data regime. Now when we increase the data size, we will start to see the major computational differences. (See Figures 6 and 7)

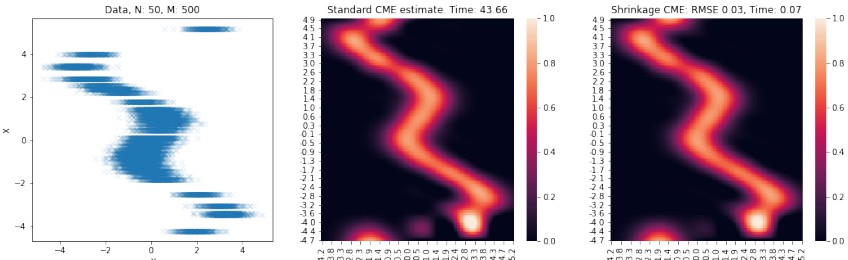

Figure 6: 50 bags with 500 samples each. (left) Data, (middle) $\hat{\mu}_{X|Y=y_i^*}(x_i^*)$ Standard CME. (right) $^S\hat{\mu}_{X|Y=y_i^*}(x_i^*)$ Shrinkage CME. With a small RMSE of $0.03$, the Shrinkage CME is approximately 600 times quicker than the standard version.

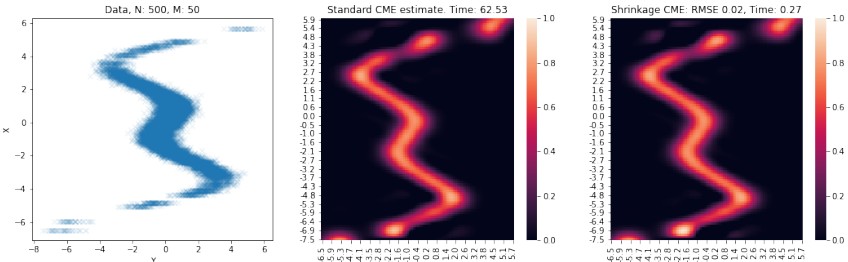

Figure 7: 500 bags with 50 samples each. (left) Data, (middle) $\hat{\mu}_{X|Y=y_i^*}(x_i^*)$ Standard CME. (right) $^S\hat{\mu}_{X|Y=y_i^*}(x_i^*)$ Shrinkage CME. Again, with a small RMSE of $0.02$, Shrinakge CME is approximately 200 times quicker than the standard CME.

# D Details on Convergence Result

In this section, we provide insights about the convergence results stated in Section 4. These results are largely based on the impactful work of Caponnetto and De Vito [36], Szabó et al. [24] and Singh et al. [25] which we modify to fit our problem setup. Each assumption that we make is adapted from a similar assumption made in those works, for which we provide intuition and a detailed justification. We start by redefining the mathematical tools introduced in these works that are necessary to state our result.

## D.1 Definitions and $\mathcal{P}_K(b, c)$ spaces

We start by providing a general definition of covariance operators over vector-valued RKHS, which will allow us to specify a class of probability distributions for our convergence result.

**Definition D.1** (Covariance operator). Let $\mathcal{W}$ a Polish space endowed with measure $\rho$, $\mathcal{G}$ a real separable Hilbert space and $K : \mathcal{W}^2 \to \mathcal{L}(\mathcal{G})$ an operator-valued kernel spanning a $\mathcal{G}$-valued RKHS $\mathcal{H}_K$.

The covariance operator of $K$ is defined as the positive trace class operator given by

$$T_K := \int_{\mathcal{Z}} K_w K_w^* d\rho(w) \in \mathcal{L}(\mathcal{H}_K) \tag{95}$$

where $\mathcal{L}(\mathcal{H}_K)$ denotes the space of bounded linear operators over $\mathcal{H}_k$.

**Definition D.2** (Power of self-adjoint Hilbert operator). Let $T$ a compact self-adjoint Hilbert space operator with spectral decomposition $T = \sum_{n=1}^{\infty} \lambda_n e_n \otimes e_n$ on $(e_n)_{n \in \mathbb{N}}$ basis of $\mathrm{Ker}(T)^\perp$. The $r^{\text{th}}$ power of $T$ is defined as $T^r = \sum_{n=1}^{\infty} \lambda_n^r e_n \otimes e_n$.

Using the covariance operator, we now introduce a general class of priors that does not assume parametric distributions, by adapting to our setup a definition originally introduced by Caponnetto and De Vito [36]. This class captures the difficulty of a regression problem in terms of two simple parameters, $b$ and $c$ [24].

**Definition D.3** ($\mathcal{P}_K(b, c)$ class). Let $\mathcal{E}_\rho : \mathcal{G}^{\mathcal{Z}} \to [0, \infty[$ an expected risk function over $\rho$ and $E_\rho = \arg\min \mathcal{E}_\rho$. Then given $b > 1$ and $c \in ]1, 2]$, we say that $\rho$ is a $\mathcal{P}_K(b, c)$ class probability measure w.r.t. $\mathcal{E}_\rho$ if

1. *Range assumption:* $\exists G \in \mathcal{H}_K$ such that $E_\rho = T_K^{\frac{c-1}{2}} \circ G$ with $\|G\|_K^2 \leq R$ for some $R \geq 0$

2. *Spectral assumption:* the eigenvalues $(\lambda_n)_{n \in \mathbb{N}}$ of $T_K$ satisfy $\alpha \leq n^b \lambda_n \leq \beta$, $\forall n \in \mathbb{N}$ for some $\beta \geq \alpha \geq 0$

The range assumption controls the functional smoothness of $E_\rho$ as larger $c$ corresponds to increased smoothness. Specifically, elements of $\mathrm{Range}(T_K^{\frac{c-1}{2}})$ admit Fourier coefficients $(\gamma_n)_{n \in \mathbb{N}}$ such that $\sum_{n=1}^{\infty} \gamma_n^2 \lambda_n^{-(c+1)} < \infty$. In the limit $c \to 1$, we obtain $\mathrm{Range}(T_K^0) = \mathrm{Range}(\mathrm{Id}_{\mathcal{H}_K}) = \mathcal{H}_K$. Since ranked eigenvalues are positive and $\lambda_n \to 0$, greater power of the covariance operator $T_K$ give rise to faster decay of the Fourier coefficients and hence smoother operators.

The spectral assumptions can be read as a polynomial decay over the eigenvalues of $T_K$. Thus, larger $b$ leads to enhanced decay $\lambda_n = \Theta(n^{-b})$ and concretely in a smaller effective input dimension.

## D.2 Complete statement of the convergence result

The following result corresponds to a detailed version of Theorem 4.2 where all the assumptions are explicitly stated. As such, its proof also constitutes the proof for Theorem 4.2.

**Theorem D.4** (Empirical DMO Convergence Rate). *Assume that*

1. *$\mathcal{X}$ and $\mathcal{Y}$ are Polish spaces, i.e. separable and completely metrizable topoligical spaces*

2. *$k$ and $\ell$ are continuous, bounded, their canonical feature maps $k_x$ and $\ell_y$ are measurable and $k$ is characteristic*

3. $\mathcal{H}_\ell$ is finite dimensional

4. $\arg\min \mathcal{E}_c \in \mathcal{H}_\Gamma$ and $\arg\min \mathcal{E}_d \in \mathcal{H}_\Gamma$

5. *The operator family* $\{\Gamma_{\mu_{X|Y=y}}\}_{y\in\mathcal{Y}}$ *is Hölder continuous with exponent* $\iota \in ]0,1]$

6. $\mathbb{P}_{XY}$ *is a* $\mathcal{P}_\Gamma(0,c')$ *class probability measure w.r.t.* $\mathcal{E}_c$ *and* $\mathbb{P}_Y$ *is a* $\mathcal{P}_\Gamma(b,c)$ *class probability measure w.r.t.* $\mathcal{E}_d$

7. $\forall g \in \mathcal{H}_\ell$, $\|g\|_\ell < \infty$ *almost surely*

*Let* $D_{\mathbb{P}_Y} = \underset{D\in\mathcal{H}_\Gamma}{\arg\min}\, \mathcal{E}_d(D)$. *Then, if we choose* $\lambda = N^{-\frac{1}{c'+1}}$ *and* $N = M^{\frac{a(c'+1)}{\iota(c'-1)}}$ *where* $a > 0$, *we have*

- *If* $a \leq \frac{b(c+1)}{bc+1}$, *then* $\mathcal{E}_d(\hat{D}_{X|Y}) - \mathcal{E}_d(D_{\mathbb{P}_Y}) = \mathcal{O}(M^{\frac{-ac}{c+1}})$ *with* $\epsilon = M^{\frac{-a}{c+1}}$

- *If* $a \geq \frac{b(c+1)}{bc+1}$, *then* $\mathcal{E}_d(\hat{D}_{X|Y}) - \mathcal{E}_d(D_{\mathbb{P}_Y}) = \mathcal{O}(M^{\frac{-bc}{bc+1}})$ *with* $\epsilon = M^{\frac{-b}{bc+1}}$

*Proof of Theorem 4.2.* The main objective here will be to rigorously verify that within our setup, the conditions in Theorem 4 from [25] are met. We reformulate from our problem perspective each of the assumptions stated by Singh et al. [25] and verify they are satisfied.

**Assumption 1** *Assume observation model* $\tilde{Z} = f(X) + \tilde{\varepsilon}$, *with* $\mathbb{E}[\tilde{\varepsilon}|Y] = 0$ *and suppose* $\mathbb{P}_{X|Y=y}$ *is not constant in* $y$.

In this work, the observation model considered is $Z = \mathbb{E}[f(X)|Y] + \varepsilon$ and the objective is to recover the underlying random variable $f(X)$ which noisy conditional expectation is observed. The latter presumes that we could bring $Z$ to $X$'s resolution. We can model it by introducing "pre-aggregation" observation model $\tilde{Z} = f(X) + \tilde{\varepsilon}$ such that $Z = \mathbb{E}[\tilde{Z}|Y]$ and $\tilde{\varepsilon}$ is a noise term at individual level satisfying $\mathbb{E}[\tilde{\varepsilon}|Y] = 0$.

**Assumption 2** $\mathcal{X}$ *and* $\mathcal{Y}$ *are Polish spaces.*

We also make this assumption.

**Assumption 3** $k$ *and* $\ell$ *are continuous and bounded, their canonical feature maps are measurable and* $k$ *is characteristic.*

We make the same assumptions. The separability of $\mathcal{X}$ and $\mathcal{Y}$ along with continuity assumptions on kernels allow to propagate separability to their associated RKHS $\mathcal{H}_k$ and $\mathcal{H}_\ell$ and to the vector-valued RKHS $\mathcal{H}_\Gamma$. Boundedness and continuity on kernels ensure the measurability of the CMO and hence that measures on $\mathcal{X}$ and $c\mathcal{Y}$ can be extended to $\mathcal{H}_k$ and $\mathcal{H}_\ell$. The assumption on $k$ being characteristic ensures that conditional mean embeddings $\mu_{X|Y=y}$ uniquely embed conditional distributions $\mathbb{P}_{X|Y=y}$ and henceforth operators over $\mathcal{H}_\ell$ are identified.

**Assumption 4** $\arg\min \mathcal{E}_c \in \mathcal{H}_\Gamma$.

This property stronger is than what the actual conditional mean operator needs to satisfy, but it is necessary to make sure the problem is well-defined. We also make this assumption.

**Assumption 5** $\mathbb{P}_{XY}$ *is a* $\mathcal{P}_\Gamma(0,c')$ *class probability measure, with* $c' \in ]1,2]$

As explained by Singh et al. [25], this is further required to bound the approximation error which we also make. Through the definition of the $\mathcal{P}_\Gamma(0,c')$ class, this hypothesis assumes the existence of a probability measure over $\mathcal{H}_k$ we denote $\mathbb{P}_{\mathcal{H}_k}$. Since $\mathcal{H}_k$ is Polish (proof below), the latter can be constructed as an extension of $\mathbb{P}_X$ over the Borel $\sigma$-algebra associated to $\mathcal{H}_k$ [48, Lemma A.3.16].

**Assumption 6** $\mathcal{H}_k$ *is a Polish space*

Since $k$ is continuous and $\mathcal{X}$ is separable, $\mathcal{H}_k$ is a separable Hilbert space which makes it Polish.

**Assumption 7** *The $\{\Gamma_{\mu_{X|Y=y}}\}_{y \in \mathcal{Y}}$ operator family is*

- Uniformly bounded in Hilbert-Schmidt norm, i.e. $\exists B > 0$ such that $\quad \forall y \in \mathcal{Y}$, $\|\Gamma_{\mu_{X|Y=y}}\|^2_{\mathrm{HS}(\mathcal{H}_\ell, \mathcal{H}_\Gamma)} \leq B$

- Hölder continuous in operator norm, i.e. $\exists L > 0, \iota \in ]0, 1]$ such that $\forall y, y' \in \mathcal{Y}, \|\Gamma_{\mu_{X|Y=y}} - \Gamma_{\mu_{X|Y=y'}}\|_{\mathcal{L}(\mathcal{H}_\ell, \mathcal{H}_\Gamma)} \leq L\|\mu_{X|Y=y} - \mu_{X|Y=y'}\|^\iota_k$

where $\mathcal{L}(\mathcal{H}_\ell, \mathcal{H}_\Gamma)$ denotes the space of bounded linear operator between $\mathcal{H}_\ell$ and $\mathcal{H}_\Gamma$.

Since we assume finite dimensionality of $\mathcal{H}_\ell$, we make a stronger assumption than the boundedness in Hilbert-Schmidt norm which we obtain as

$$\|\Gamma_{\mu_{X|Y=y}}\|^2_{\mathrm{HS}(\mathcal{H}_\ell, \mathcal{H}_\Gamma)} = \mathrm{tr}\left(\Gamma(\mu_{X|Y=y}, \mu_{X|Y=y})\right) \tag{96}$$

$$= \mathrm{tr}\left(\langle \mu_{X|Y=y}, \mu_{X|Y=y}\rangle_k \, \mathrm{Id}_{\mathcal{H}_\ell}\right) \tag{97}$$

$$= q(y, y)\,\mathrm{tr}\left(\mathrm{Id}_{\mathcal{H}_\ell}\right) < \infty. \tag{98}$$

Hölder continuity is a mild assumption commonly satisfied as stated in [24].

**Assumption 8** $\arg\min \mathcal{E}_{\mathrm{d}} \in \mathcal{H}_\Gamma$ *and $\mathcal{H}_\ell$ is a space of bounded functions almost surely*

We assume that the true minimiser of $\mathcal{E}_{\mathrm{d}}$ is in $\mathcal{H}_\Gamma$ to have a well-defined problem. The second assumption here is expressed in terms of probability measure $\mathbb{P}_{\mathcal{H}_\ell}$ over $\mathcal{H}_\ell$. We do also assume that there exists $B > 0$ such that $\forall g \in \mathcal{H}_\ell, \|g\|_\ell < B \; \mathbb{P}_{\mathcal{H}_\ell}-$ almost surely.

**Assumption 9** $\mathbb{P}_Y$ *is a $\mathcal{P}_\Gamma(b, c)$ class probability measure, with $b > 1$ and $c \in ]1, 2]$*

This last hypothesis is not required per se to obtain a bound on the excess error of regularized estimate $\hat{D}_{X|Y}$. However, it allows to simplify the bounds and state them in terms of parameters $b$ and $c$ which characterize efficient input size and functional smoothness respectively.

Furthermore, a premise to this assumption is the existence of a probability measure over $\mathcal{H}_\ell$ that we denote $\mathbb{P}_{\mathcal{H}_\ell}$. Since $\ell$ is continuous and $\mathcal{Y}$ separable, it makes $\mathcal{H}_\ell$ a separable and thus Polish. We can then construct $\mathbb{P}_{\mathcal{H}_\ell}$ by extension of $\mathbb{P}_Y$ [48, Lemma A.3.16] $\qquad\qquad\qquad\square$

This theorem underlines a trade-off between the computational and statistical efficiency w.r.t. the datasets cardinalities $N = |\mathcal{D}_1|$ and $M = |\mathcal{D}_2|$ and the problem difficulty $(b, c, c')$.

For $a \leq \frac{b(c+1)}{bc+1}$, smaller $a$ means less samples from $\mathcal{D}_1$ at fixed $M$ and thus computational savings. But it also hampers convergence, resulting in reduced statistical efficiency. At $a = \frac{b(c+1)}{bc+1} < 2$, convergence rate is a minimax computational-statistical efficiency optimal, i.e. convergence rate is optimal with smallest possible $M$. We note that at this optimal, $N > M$ and hence we require less samples from $\mathcal{D}_2$. $a \geq \frac{b(c+1)}{bc+1}$ does not improve the convergence rate but increases the size of $\mathcal{D}_1$ and hence the computational cost it bears.

We also note that larger Hölder exponents $\iota$, which translates in smoother kernels, leads to reduced $N$. Similarly, since $c' \mapsto \frac{c'+1}{c'-1}$ and $c \mapsto \frac{b(c+1)}{bc+1}$ are strictly decreasing functions over $]1, 2]$, stronger range assumptions regularity which means smoother operators reduces the number of sample needed from $\mathcal{D}_1$ to achieve minimax optimality. Smoother problems do hence require fewer samples.

Larger spectral decay exponent $b$ translate here in requiring more samples to reach minimax optimality and undermines optimal convergence rate. Hence problems with smaller effective input dimension are harder to solve and require more samples and iterations.

# E   Additional Experimental Results

## E.1   Swiss Roll Experiment

### E.1.1   Statistical significance table

Table 3: p-values from a two-tailed Wilcoxon signed-rank test between all pairs of methods for the test RMSE of the swiss-roll experiment with a direct and indirect matching setup. The null hypothesis is that scores samples come from the same distribution. We only present the lower triangular matrix of the table for clarity of reading.

| Matching | | CMP | BAGG-GP | VARCMP | VBAGG | GPR | S-CMP |
|---|---|---|---|---|---|---|---|
| Direct | CMP | - | - | - | - | - | - |
| | BAGG-GP | 0.00006 | - | - | - | - | - |
| | VARCMP | 0.00008 | 0.00006 | - | - | - | - |
| | VBAGG | 0.00006 | 0.00006 | 0.005723 | - | - | - |
| | GPR | 0.00006 | 0.00006 | 0.00006 | 0.00006 | - | - |
| | S-CMP | 0.00006 | 0.00006 | 0.000477 | 0.014269 | 0.00006 | - |
| Indirect | CMP | - | - | - | - | - | - |
| | BAGG-GP | 0.011129 | - | - | - | - | - |
| | VARCMP | 0.001944 | 0.015240 | - | - | - | - |
| | VBAGG | 0.000089 | 0.047858 | 0.000089 | - | - | - |
| | GPR | 0.025094 | 0.047858 | 0.047858 | 0.851925 | - | - |
| | S-CMP | 0.000089 | 0.002821 | 0.000089 | 0.000140 | 0.052222 | - |

### E.1.2   Compute and Resources Specifications

Computations for all experiments were carried out on an internal cluster. We used a single GeForce GTX 1080 Ti GPU to speed up computations and conduct each experiment with multiple initialisation seeds. We underline however that the experiment does not require GPU acceleration and can be performed on CPU in a timely manner.

### E.2 CMP with high-resolution noise observation model

#### E.2.1 Deconditional posterior with high-resolution noise

Beyond observation noise on the aggregate observations $\tilde{\mathbf{z}}$ as introduced in Section **??**, it is natural to also consider observing noise at the high-resolution level, i.e. noises placed on $f$ level directly in addition to the one $g$ at aggregate level. Let $\xi \sim \mathcal{GP}(0, \delta)$ the zero-mean Gaussian process with covariance function

$$\delta : \left| \begin{array}{ccl} \mathcal{X} \times \mathcal{X} & \longrightarrow & \mathbb{R} \\ (x, x') & \longmapsto & \begin{cases} 1 \text{ if } x = x' \\ 0 \text{ else} \end{cases} \end{array} \right. . \tag{99}$$

By incorporating this gaussian noise process in the integrand, we can replace the definition of the CMP by

$$g(y) = \int_{\mathcal{X}} (f(x) + \varsigma\xi(x)) \, \mathrm{d}\mathbb{P}_{X|Y=y}, \quad \forall y \in \mathcal{Y}, \tag{100}$$

where $\varsigma > 0$ is the high-resolution noise standard deviation parameter. Essentially, this amounts to consider a contaminated covariance for the HR observation process. This covariance is defined as

$$k^\varsigma : \left| \begin{array}{ccl} \mathcal{X} \times \mathcal{X} & \longrightarrow & \mathbb{R} \\ (x, x') & \longmapsto & k(x, x') + \varsigma^2\delta(x, x') \end{array} \right. . \tag{101}$$

Provided the same regularity assumptions as in Proposition 3.2, the covariance of the CMP becomes $q(y, y') = \mathbb{E}[k^\varsigma(X, X')|Y = y, Y' = y']$ — the mean and cross-covariance terms are not affected. Similarly be written in terms of conditional mean embeddings, but using as an integrand for the CMEs the canonical feature maps induced by $k^\varsigma$, i.e. $\mu^\varsigma_{X|Y=y} := \mathbb{E}[k^\varsigma(X, \cdot)|Y = y]$ for any $y \in \mathcal{Y}$. Critically, this is reflected in the expression of the empirical CMP covariance which writes

$$\hat{q}(y, y') = \ell(y, \mathbf{y})(\mathbf{L_{yy}} + N\lambda\mathbf{I}_N)^{-1}(\mathbf{K_{xx}} + \varsigma^2\mathbf{I}_N)(\mathbf{L_{yy}} + N\lambda\mathbf{I}_N)^{-1}\ell(\mathbf{y}, y') \tag{102}$$

thus, yielding matrix form

$$\hat{\mathbf{Q}}_{\tilde{\mathbf{y}}\tilde{\mathbf{y}}} := \hat{q}(\tilde{\mathbf{y}}, \tilde{\mathbf{y}}) \tag{103}$$

$$= \mathbf{L_{\tilde{y}y}}(\mathbf{L_{yy}} + N\lambda\mathbf{I}_N)^{-1}(\mathbf{K_{xx}} + \varsigma^2\mathbf{I}_N)(\mathbf{L_{yy}} + N\lambda\mathbf{I}_N)^{-1}\mathbf{L_{y\tilde{y}}} \tag{104}$$

$$= \mathbf{A}^\top(\mathbf{K_{xx}} + \varsigma^2\mathbf{I}_N)\mathbf{A}. \tag{105}$$

which can readily be used in (8) and (9) to compute the deconditional posterior.

This high-resolution noise term introduces an additional regularization to the model that helps preventing degeneracy of the deconditional posterior covariance. Indeed, we have

$$\hat{k}_{\mathrm{d}}(\mathbf{x}, \mathbf{x}) = \mathbf{K_{xx}} - \mathbf{K_{xx}}\mathbf{A}(\hat{\mathbf{Q}}_{\tilde{\mathbf{y}}\tilde{\mathbf{y}}} + \sigma^2\mathbf{I}_M)^{-1}\mathbf{A}^\top\mathbf{K_{xx}} \tag{106}$$

$$= \mathbf{K_{xx}} - \mathbf{K_{xx}}\mathbf{A}(\mathbf{A}^\top(\mathbf{K_{xx}} + \varsigma^2\mathbf{I}_N)\mathbf{A} + \sigma^2\mathbf{I}_M)^{-1}\mathbf{A}^\top\mathbf{K_{xx}} \tag{107}$$

$$= \mathbf{K_{xx}} - \mathbf{K_{xx}}(\mathbf{A}\mathbf{A}^\top(\mathbf{K_{xx}} + \varsigma^2\mathbf{I}_N) + \sigma^2\mathbf{I}_M)^{-1}(\mathbf{A}\mathbf{A}^\top\mathbf{K_{xx}}). \tag{108}$$

where on the last line we have used the Woodburry identity. We can see that when $\sigma = \varsigma = 0$, (108) degenerates to 0. The aggregate observation model noise $\sigma$ provides a first layer of regularization at low-resolution. The high-resolution noise $\varsigma$ supplements it, making for a more stable numerical compuation for the empirical covariance matrix.

#### E.2.2 Variational deconditional posterior with high-resolution noise

The high-resolution noise observation process can also be incorporated into the variational derivation to obtain a slightly different ELBO objective. We have

$$p(\tilde{\mathbf{z}}|\mathbf{f}) = \mathcal{N}(\tilde{\mathbf{z}}|\mathbf{\Upsilon}^\top\mathbf{K_{xx}}^{-1}\mathbf{f}, \ \mathbf{Q}_{\tilde{\mathbf{y}}\tilde{\mathbf{y}}} + \sigma^2\mathbf{I}_M - \mathbf{\Upsilon}^\top\mathbf{K_{xx}}^{-1}\mathbf{\Upsilon}) \tag{109}$$

$$= \mathcal{N}(\tilde{\mathbf{z}}|\mathbf{A}\mathbf{f}, \ \mathbf{A}^\top(\mathbf{K_{xx}} + \varsigma^2\mathbf{I}_N)\mathbf{A} + \sigma^2\mathbf{I}_M - \mathbf{A}^\top\mathbf{K_{xx}}\mathbf{A}) \tag{110}$$

$$= \mathcal{N}(\tilde{\mathbf{z}}|\mathbf{A}\mathbf{f}, \ \varsigma^2\mathbf{A}^\top\mathbf{A} + \sigma^2\mathbf{I}_M) \tag{111}$$

The expected loglikelihood with respect to the variational posterior hence writes

$$\mathbb{E}_{q(\mathbf{f})}[p(\tilde{\mathbf{z}}|\mathbf{f})] = -\frac{M}{2}\log(2\pi) - \frac{1}{2}\log\det(\varsigma^2\mathbf{A}^\top\mathbf{A} + \sigma^2\mathbf{I}_M) \tag{112}$$

$$-\frac{1}{2}\mathbb{E}_{q(\mathbf{f})}\left[(\tilde{\mathbf{z}} - \mathbf{A}^\top\mathbf{f})^\top(\varsigma^2\mathbf{A}^\top\mathbf{A} + \sigma^2\mathbf{I}_M)^{-1}(\tilde{\mathbf{z}} - \mathbf{A}^\top\mathbf{f})\right] \tag{113}$$

With a derivation similar to the one proposed in Appendix B, the expected loglikelihood can be expressed in terms of the posterior variational parameters as

$$\mathbb{E}_{q(\mathbf{f})}[p(\tilde{\mathbf{z}}|\mathbf{f})] = -\frac{M}{2}\log(2\pi) - \frac{1}{2}\log\det(\varsigma^2\mathbf{A}^\top\mathbf{A} + \sigma^2\mathbf{I}_M) \tag{114}$$

$$-\frac{1}{2}(\tilde{\mathbf{z}} - \mathbf{A}^\top\bar{\boldsymbol{\eta}})^\top(\varsigma^2\mathbf{A}^\top\mathbf{A} + \sigma^2\mathbf{I}_M)^{-1}(\tilde{\mathbf{z}} - \mathbf{A}^\top\bar{\boldsymbol{\eta}}) \tag{115}$$

$$-\frac{1}{2}\operatorname{tr}\left((\varsigma^2\mathbf{A}^\top\mathbf{A} + \sigma^2\mathbf{I}_M)^{-1}\mathbf{A}^\top\bar{\boldsymbol{\Sigma}}\mathbf{A}\right) \tag{116}$$

In particular, the last term can be rearranged into $\operatorname{tr}\left(\bar{\boldsymbol{\Sigma}}^{1/2}\mathbf{A}(\varsigma^2\mathbf{A}^\top\mathbf{A} + \sigma^2\mathbf{I}_M)^{-1}\mathbf{A}^\top\bar{\boldsymbol{\Sigma}}^{1/2}\right)$ which can efficiently be computed as an inverse quadratic form [38].

### E.3 Mediated downscaling of atmospheric temperature

### E.3.1 Map visualization of atmospheric fields dataset

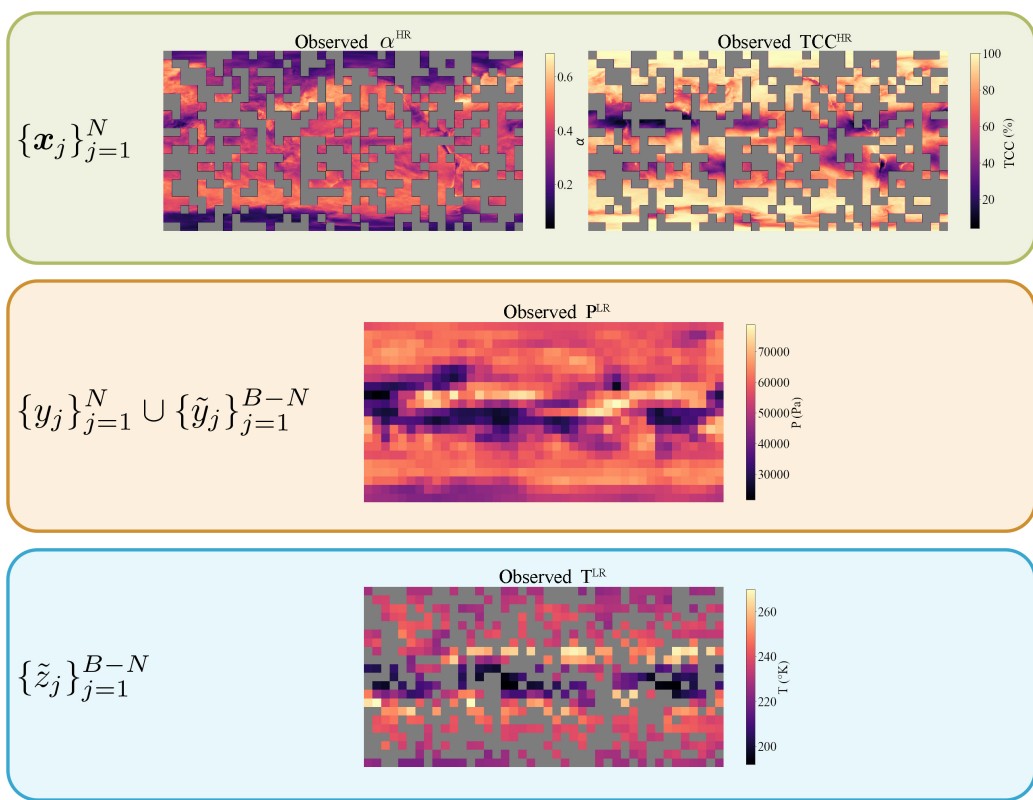

Figure 8: Map visualization of the dataset used in the mediated downscaling experiment (for one random seed); **Top:** Bags of high-resolution albedo $\alpha^{\text{HR}}$ and total cloud cover $\text{TCC}^{\text{HR}}$ pixels which are observed in $\mathcal{D}_1$ — each "coarse pixel" delineates a bag of HR pixels; **Middle:** Low-resolution pressure field $\text{P}^{\text{LR}}$ which is observed everywhere and plays the role of mediating variable; **Bottom:** Low-resolution temperature field $\text{T}^{\text{LR}}$ pixels which are observed in $\mathcal{D}_2$ and that we want to downscale; grey pixels are unobserved; the grey layer on HR covariates maps (top) is the exact complementary of the grey layer on the observed $\text{T}^{\text{LR}}$ map (bottom).

## E.3.2 Downscaling prediction maps

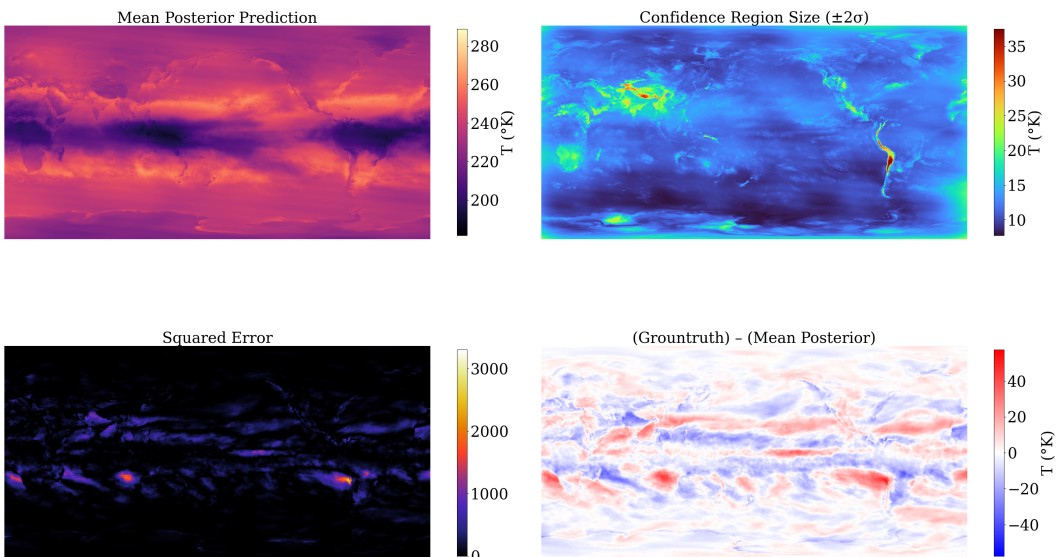

Figure 9: Predicted downscaled atmospheric temperature field with VARGPR; **Top-Left:** Posterior mean; **Top-Right:** 95% confidence region size, i.e. 2 standard deviation of the posterior; **Bottom-Left:** Squared difference with unobserved groundtruth $T^{HR}$; **Bottom-Right:** Difference between unobserved groundtruth $T^{HR}$ and the posterior mean.

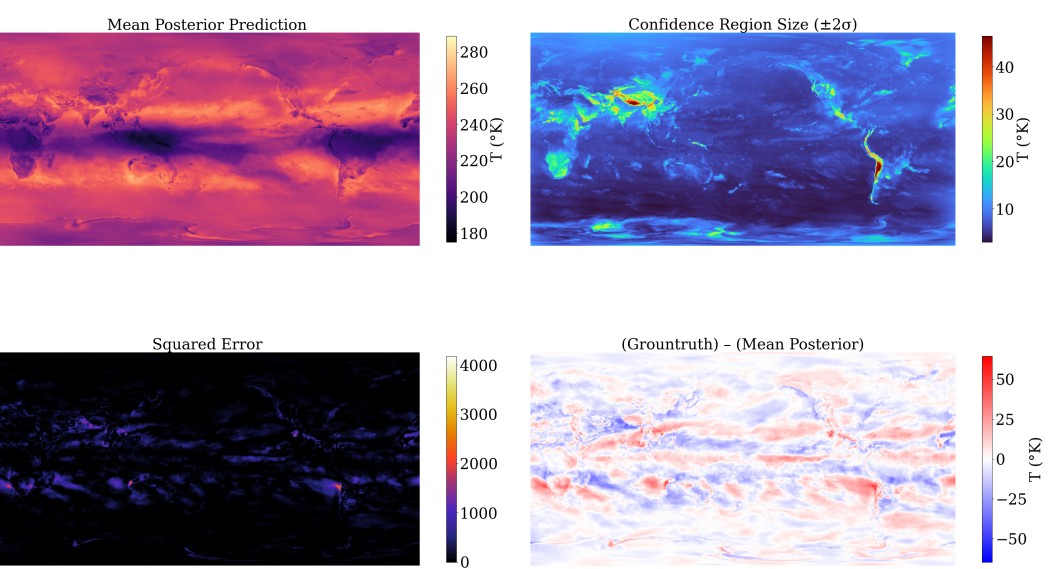

Figure 10: Predicted downscaled atmospheric temperature field with VBAGG; **Top-Left:** Posterior mean; **Top-Right:** 95% confidence region size, i.e. 2 standard deviation of the posterior; **Bottom-Left:** Squared difference with unobserved groundtruth $T^{HR}$; **Bottom-Right:** Difference between unobserved groundtruth $T^{HR}$ and the posterior mean.

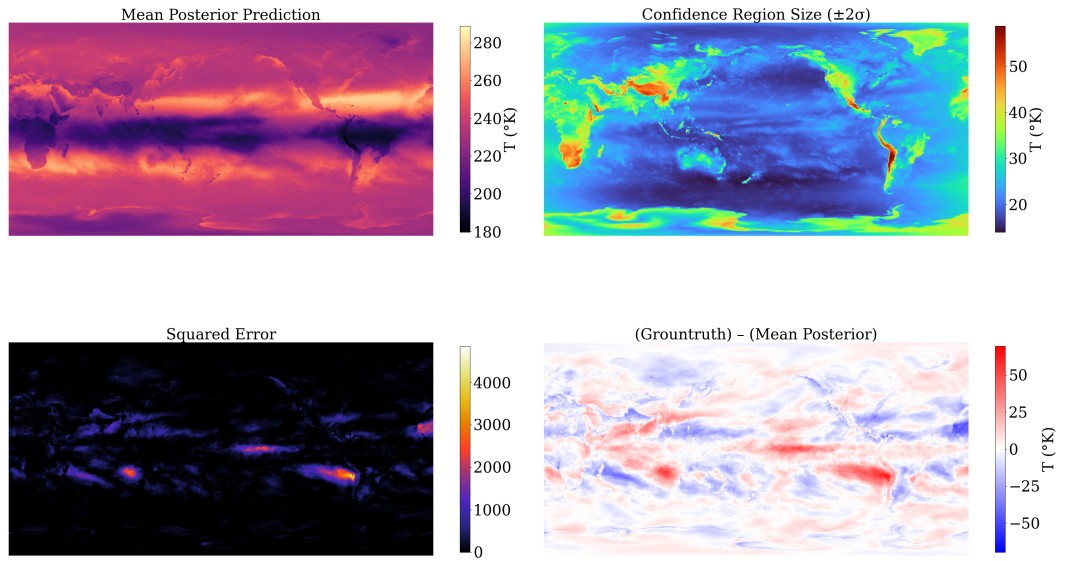

Figure 11: Predicted downscaled atmospheric temperature field with VARCMP; **Top-Left:** Posterior mean; **Top-Right:** 95% confidence region size, i.e. 2 standard deviation of the posterior; **Bottom-Left:** Squared difference with unobserved groundtruth $T^{HR}$; **Bottom-Right:** Difference between unobserved groundtruth $T^{HR}$ and the posterior mean.

### E.3.3 Statistical significance table

Table 4: p-values from a two-tailed Wilcoxon signed-rank test between all pairs of methods for the evaluation scores on the mediated statistical downscaling experiment. The null hypothesis is that scores samples come from the same distribution. As before, we only present the lower-traingular table for clarity of reading.

| Metric | | VARCMP | VBAGG | VARGPR |
|---|---|---|---|---|
| RMSE | VARCMP | - | - | - |
| | VBAGG | 0.005062 | - | - |
| | VARGPR | 0.006910 | 0.046853 | - |
| MAE | VARCMP | - | - | - |
| | VBAGG | 0.005062 | - | - |
| | VARGPR | 0.059336 | 0.006910 | - |
| CORR | VARCMP | - | - | - |
| | VBAGG | 0.005062 | - | - |
| | VARGPR | 0.016605 | 0.028417 | - |
| SSIM | VARCMP | - | - | - |
| | VBAGG | 0.005062 | - | - |
| | VARGPR | 0.959354 | 0.005062 | - |

### E.3.4 Compute and Resources Specifications

Computations for all experiments were carried out on an internal cluster. We used a single GeForce GTX 1080 Ti GPU to speed up computations and conduct each experiment with multiple initialisation seeds.