# OpenReview forum: "Deconditional Downscaling with Gaussian Processes"
_NeurIPS.cc/2021/Conference — NeurIPS 2021 Poster_

### Official Review · Reviewer_GawS · 2021-07-09

**Rating:** 6
**Confidence:** 2

**Summary:**

The paper proposes a novel Gaussian process approach for settings in which we have two datasets, D_1={(x_i, y_i)}_i=1 ^N and D_2=((y_i,z_i))_i=1^M  that are linked by the mediating variable Y. The datasets are not matched, and we want to learn a function from X to Z. In contrast to existing work, the algorithm generalizes to settings in which the resolution of the two datasets differ.

**Limitations And Societal Impact:**

The societal impact paragraph is sufficient.

**Main Review:**

The paper comes along with a theoretical analysis of the proposed approach and a careful experimental validation. In addition, the paper is accomodated by a github repository that looks well documented and easy to run. While the paper seems to be theoretically sound, I found parts of it quiet confusing. However, this might also be attributed to my rather scarce background knowledge on conditional mean embeddings.

The title and abstract of the paper suggest that the main innovation of the paper is a new GP formulation that allows for different resolutions. However, for large parts of the paper, this seems to not be very relevant. In fact, the down-scaling only enters in Section 3.3 and in the experiments and is not even part of the main notation.

The major contribution of the paper seems to me is more the definition of the conditional mean process which allows for mediating variables in general. Interestingly,  one arrives at a GP formulation including analytical solutions for mean and covariance. This was surprising to me because chaining two Gaussian processes together and marginalizing out the intermediate variable leads in general to non-Gaussian distributions.

The posterior mean recovers a solution that has been published previously using the deconditional mean embedding [19].
For ease of readibility, I would suggest to move the complete deconditional mean embedding discussion in Section 4 and not scattered between Section 1, Section 2.2 and Section 4. It is not strictly needed for understanding the conditional mean process and distracts from it.

I also wonder if the paper would not be better split into two separate contributions - one with the general theoretical framework and one with the specific down-scaling application in mind. For me, the two parts seem to be a bit disconnected.

In the experimental section, I would suggest to add a second metric that takes the uncertainty estimates of the GP into account. After all, this is one of the main benefits of the GP-based method. It would also be interesting to compare the uncertainty estimates between the proposed solution and the GP in [19] for the special case that the granularity levels are the same on both datasets.

**Time Spent Reviewing:**

4

---

> ### Author Response · Authors · 2021-08-10
> **Reply to Reviewer GawS**
>
> *“The title and abstract of the paper suggest that the main innovation of the paper is a new GP formulation that allows for different resolutions. However, for large parts of the paper, this seems to not be very relevant. In fact, the down-scaling only enters in Section 3.3 and in the experiments and is not even part of the main notation.”*
>
> A: We thank the reviewer for raising this point and we apologize for the confusion this might have caused. While the problem setup is introduced in terms of bagged data in Section 1 and 2, we presented the general framework using non-bagged notations in Section 3 and 4 as we believe our general model and theory subsumes the multiresolution setting (as pointed out by Reviewer wwEN ). We will ensure this is further clarified in the camera-ready version.
>
> ------------------------------------------------------------------------------------------------------------------------------------------------------
>
> *“The major contribution of the paper seems to me is more the definition of the conditional mean process which allows for mediating variables in general.”*
>
> A: We believe that Conditional Mean Processes (CMP) are indeed an interesting contribution, but we believe they constitute a mild novelty to our work as their properties and characterisations are simple to derive. We see our major contributions as follows:
>
> We believe that being able to phrase the deconditioning framework from Hsu and Ramos (2019) as a generalisation of VBAGG-like spatial disaggregation models bridges two disparate research topics in a novel and non-trivial way.
> We propose a statistical downscaling method in the context where a mediating variable is observed, which is commonly the case when working with climate data.
> We established deconditional mean operators as vector-valued regressors which highlight its interpretation as a reconstruction operator and provide guarantees on the convergence rate of the operators using the setup and results from the two-staged regression literature.
>
> ------------------------------------------------------------------------------------------------------------------------------------------------------
>
> *“Interestingly, one arrives at a GP formulation including analytical solutions for mean and covariance. This was surprising to me because chaining two Gaussian processes together and marginalizing out the intermediate variable leads in general to non-Gaussian distributions.”*
>
> A: We would like to clarify with the reviewer what they mean by “chaining GPs”. In our work, we did not chain GPs but instead, we explicitly characterise the cross-covariance relationship between the integrated GP (g) and the latent GP (f) and use Gaussian conditioning to recover the posterior of the latent GP (f). In fact, this is strongly related to the ideas developed in the inter-domain GP literature. Please see [30] for more details.
>
> ------------------------------------------------------------------------------------------------------------------------------------------------------
>
> *“The posterior mean recovers a solution that has been published previously using the deconditional mean embedding [19]. For ease of readibility, I would suggest to move the complete deconditional mean embedding discussion in Section 4 and not scattered between Section 1, Section 2.2 and Section 4. It is not strictly needed for understanding the conditional mean process and distracts from it. ... I also wonder if the paper would not be better split into two separate contributions - one with the general theoretical framework and one with the specific down-scaling application in mind. For me, the two parts seem to be a bit disconnected.”*
>
> A: We apologize if the paper structure has been misleading. As underlined above, CMPs are in this work only introduced as a means to the deconditional posterior. The properties of CMPs are key to recover the deconditioning solution and hence worth being explicitly stated in Section 3.1. These properties are however fairly simple and we do not believe however that CMPs are a contribution worth dedicating an entire section. Deconditioning, on the contrary, is a core focus of this work and not a mere discussion. The paper structure is intended to set the focus on the derivation of the deconditional posterior in Section 3, which we believe to be one of the main contributions.
>
> ------------------------------------------------------------------------------------------------------------------------------------------------------
>
> *“In the experimental section, I would suggest to add a second metric that takes the uncertainty estimates of the GP into account. After all, this is one of the main benefits of the GP-based method. It would also be interesting to compare the uncertainty estimates between the proposed solution and the GP in [19] for the special case that the granularity levels are the same on both datasets.”*
>
> A: These are indeed good suggestions and we thank the reviewer for that. Given the amount of content and contributions we are trying to communicate, we preferred not to overload section 5. Using a probabilistic metric and comparing models across different granularity levels to benchmark methods are a very sound addition to the paper. In fact, we are at the moment considering a calibration study of the uncertainty estimates of the deconditional posterior, i.e comparing the credible interval’s true coverage probability to the nominal coverage probability and plan to include this additional experiment in the camera-ready version.

---

> > ### Comment · Reviewer_GawS · 2021-09-01
> > **Reply to Authors**
> >
> > I thank the authors for the detailed answers to all reviews. As written in my initial review, I was confused when reviewing the paper how the different parts fit together and if a better structure exists in order to improve readability. While I appreciate the
> > feedback of the authors and in particular the new introduction, I would need to re-assess the complete manuscript after restructuring/rewriting in order to increase my score.

---

### Official Review · Reviewer_sfJ5 · 2021-07-10

**Rating:** 6
**Confidence:** 3

**Summary:**

This paper presents a GP model for refining coarse-grained spatial data, which can be used for handling aggregated data that are unmatched spatially and temporally. The authors also develop the two-stage regression algorithm for downscaling. The effectiveness of the proposed model is demonstrated using synthetic and real-world datasets.

**Limitations And Societal Impact:**

Please see comments (1), (2), and (3) in my main review.

**Main Review:**

The task of recovering spatial fields from aggregated data is interesting and important. This paper is well-written and technically sound. The use of Conditional and Deconditional Kernel Mean Embeddings to the downscaling problem is novel. The experiments on atmospheric temperature data are encouraging. Accordingly, my opinion tends to accept; but there are some concerns about technical issues.

The detailed comments are as follows.

(1) It seems that the input of the proposed method is limited to data aggregated at grid cells. But, in my understanding, the proposed method allows data aggregated over regions that have various shapes; is that right? If that is the case, it would be helpful to clarify it; this idea may be similar to those of [e.g., 10, R1].

Actually, the spatially aggregated data (e.g., disease incidence [7]) are often associated with irregularly shaped regions such as districts. As in [10, R1], can the proposed model handle the effects from the region's size and shape? That could be critical for estimating prediction uncertainty.

(2) I support you in that GP-based modeling is useful, but it would be better to mention the recently published downscaling methods based on DNN [e.g., R2] and clarify the advantages of the proposed approach.

(3) In [6], learning from aggregated data is more generally formulated. Can the authors consider the extension of the proposed method to other machine learning tasks such as classification and ordinal regression?

(4) Could you please elaborate on the reason why the predictive performance of BAGG-GP and VBAGG is worse than vanilla GP?

[R1] M. T. Smith, M. A. Álvarez, and N. D. Lawrence. Gaussian process regression for binned data. In arXiv e-prints, 2018.

[R2] Vandal, T.; Kodra, E.; Ganguly, S.; Michaelis, A.; Nemani, R.; and Ganguly, A. R. 2017. DeepSD: Generating high resolution climate change projections through single image super-resolution. In KDD, 1663–1672.

**Time Spent Reviewing:**

12 hours

---

> ### Author Response · Authors · 2021-08-10
> **Reply to Reviewer sfJ5**
>
> _“(1) It seems that the input of the proposed method is limited to data aggregated at grid cells. But, in my understanding, the proposed method allows data aggregated over regions that have various shapes; is that right? If that is the case, it would be helpful to clarify it; this idea may be similar to those of [e.g., 10, R1].
> Actually, the spatially aggregated data (e.g. disease incidence [7]) are often associated with irregularly shaped regions such as districts. As in [10, R1], can the proposed model handle the effects from the region's size and shape? That could be critical for estimating prediction uncertainty.”_
>
> A: This is absolutely right, we thank the reviewer for raising this point. As pointed out by the reviewer, this method does generalize to regions having various shapes – since the formalism introduced in Section 2 and 3 does not assume any grid structure for the data – and could for example be applied to the irregularly shaped aerial data from [10]. In fact, the swiss roll experiment from Section 5.1 also does not assume any grid structure for the data and our method is still applicable. We shall clarify this point in the revised version.
>
> ------------------------------------------------------------------------------------------------------------------------------------------------------
>
> _“(2) I support you in that GP-based modeling is useful, but it would be better to mention the recently published downscaling methods based on DNN [e.g., R2] and clarify the advantages of the proposed approach.”_
>
> A: This is a very fair point, thank you for bringing this up. We agree that a discussion about the differences from DNN-based methods will be useful. DNN-based downscaling approaches [R2, R3, R4, R5] most generally assume availability of training data, i.e. that both high-resolution and low-resolution matched observations are available, and in large amounts. Then, they simply follow a supervised learning setting in learning a map between different resolutions. Our method never assumes that high-resolution targets are observed. Only low-resolution samples of the targets ($\tilde{z}$) are observed, and it is through high-resolution covariates ($x$) that the high-resolution versions of the targets are inferred. DNN-based approaches are hence not applicable in our problem and observation setting.
>
>
> [R2] Vandal, T.; Kodra, E.; Ganguly, S.; Michaelis, A.; Nemani, R.; and Ganguly, A. R. 2017. DeepSD: Generating high resolution climate change projections through single image super-resolution. In KDD, 1663–1672.
>
>
> [R3] B. Groenke, L. Madaus, and C. Monteleoni, “ClimAlign: Unsupervised statistical downscaling of climate variables via normalizing flows,” arXiv, 2020.
>
>
> [R4] A. Vaughan, W. Tebbutt, J. S. Hosking, and R. E. Turner, “Convolutional conditional neural processes for local climate downscaling,” pp. 1–26, 2021.
>
>
> [R5] M. Deudon et al., “HighRes-net: Recursive Fusion for Multi-Frame Super-Resolution of Satellite Imagery,” 2020.
>
> ------------------------------------------------------------------------------------------------------------------------------------------------------
>
> _“(3) In [6], learning from aggregated data is more generally formulated. Can the authors consider the extension of the proposed method to other machine learning tasks such as classification and ordinal regression?”_
>
> A: The generalisation to observation models beyond the regression setting is an exciting research direction that we are actively investigating. We thank the reviewer for pointing this out. Since the resulting Conditional Mean Process (CMP) may not be a GP anymore for a classification task for example, one may not readily recover the deconditional posterior using Gaussian conditioning. It should, however, be possible to construct tractable approximations of the posterior using variational techniques such as those developed in [Law et al. 2018] for the case of Poisson likelihood and in our work for the purposes of scalability. This will be an important future research direction
>
> ------------------------------------------------------------------------------------------------------------------------------------------------------
>
> _“(4) Could you please elaborate on the reason why the predictive performance of BAGG-GP and VBAGG is worse than vanilla GP?”_
>
> A: As reported in Table 3 from Appendix E.1.1, the p-value for the comparison between BAGG-GP and GPR is borderline (0.47858), so it is unclear how much this significance should be relied upon drawing conclusions. We emphasise that both GPR and BAGG-GP methods are in fact ill-suited for the unmatched case and perform poorly in comparison to the proposed method.

---

### Official Review · Reviewer_szhM · 2021-07-14

**Rating:** 5
**Confidence:** 2

**Summary:**

The paper considers the problem of learning a function
$$f:\mathcal{X}\to \mathbb{R}$$ from the relationship
$$g(y) = \int_x f(x) \mathbb{P}[x|y]dx$$
where the training data consists of two datasets. The first set is  $\mathcal{D}_1 = ( x^b_i, y_i )$ where each $x^b_i$ is a bag of points in $\mathcal{X}$ and $y_i$ is a single point in a space $\mathcal{Y}$, these are empirical samples from some distribution $\mathbb{P}[x|y]\mathbb{P}[y]$. The second dataset is of $\mathcal{D}_2 = (\tilde y_j, \tilde z_i)$ with $\tilde y_j \in \mathcal{Y}$ and $\tilde z_i \in \mathbb{R}$ is a set of input-output pairs to from $\tilde z_i = g(\tilde y_i)$.

Learning models for $f(x)$ from aggregate data, (input bag, output) pairs e.g. $( x^b_i, z_i )$ has been studied, similarly the use of  there are intermediate covariates $y_i \in \mathcal{Y}$ with a dataset of the form $( x^b_i, y_i, z_i )$ has been studied. As I undertsand, the novel constributino of this work is the extension to the case where there are two datasets, $( x^b_i, y_i)$ and $(\tilde y_j, \tilde z_j )$ that both contain points from the domain $\mathcal{Y}$ yet they are not the same points, they are mismatched.

The paper proposes a "conditional mean process", a Gaussian process model that interpolates aggregated data as well as the marginalization operater in order to infer the un-aggregated latent function under this above problem setting.



**Ethical Concerns:**

I have no ethical concerns.

**Limitations And Societal Impact:**

The authors do not appear to explicitly raise possible failure modes, below is one example.

  - if $l_y$ is poorly chosen, this may cause the method to perform worse than VBAGG in matched data setting and worse than naive methods (like the imputation method on line 338) in the unmatched data setting.

**Main Review:**

I have two major concerns, the overall writing and the novelty of the proposed conditional mean process.

Regarding the Conditional Mean Process

  -  the integral of a Gaussian process is another Gaussian process is a standard result (eg convolution of a GP is another GP), I feel I have seen variations of equation (3) in multiple previous works [1] [2].

  - I believe the following "deconditioning" step is standard (as rightly acknowledged to be Gaussian conditioning) and equations (5), (6) are textbook results.

  - the sparse inducing point variational approach was applied in Hsu and Ramos

  - I believe the main practical novelty mainly lies in using the kernel $l_y$ as a means to model $\mathbb{P}[X|Y]$ away from the training data thereby bridging the gap between the unmatched $y_i$ and $\tilde y_j$, I think this is really cool and the inclusion of VBAGG baseline (where $l_y$ is effectively "switched off") demonstrates the efficacy. However this seems rather unsurprising and rather incremental.

[1] https://arxiv.org/abs/1902.07908

[2] https://ieeexplore.ieee.org/abstract/document/9383978


Overall I found the paper initially rather difficult to read, I felt the mathematical statements were very dense (mainly Section 2.2) with minimal explanation and required multiple passes to follow (I am largely familiar with the notation though not an expert). I had to read the paper multiple times to first acquire an intuitive understanding of the problem definition, I felt there were multiple subtle ambiguities that threw me off course and minor clarifications would have made the paper much easier to read.

Here are some of my personal writing nitpicks.
 -  the first dataset is is introduced as $(x, y)$. This would traditionally imply that $x$ is predicting $y$, i.e. modelling $\mathbb{P}[Y|X]$, Figure 1 seems to imply this as well, however $y$ is an "input" for the output $x^b$, it is used to Monte Carlo approximate $\mathbb{P}[X|Y]$ instead. (I felt the introductory paragraph of Hsu and Ramos [19] was far easier to follow), perhaps using $y$ as function outputs (like [3]) as in  and $z$ (or anything else) as the "middle man" variables.

 - $z = \mathbb{E}[f(X)|Y] + \epsilon$ does not explicitly state what the expectation is over, though it slowly becomes clear that is is $X$.

  - the word "conditional" is used very frequently and I believe it is referring to the _marginal_ over $X$ using $\mathbb{P}[X|Y]$ (which happens to be a conditional distribution although $Y$ could be any parameter). Again upon first reading this led to a lot of ambiguity for me. (I would personally find "de-marginalizing" or "de-aggregation" to be much more intuitive than "de-conditioning")

  - given the two datasets (x, y), (y, z) and the "two stage regression" statements, I was partly expecting deep Gaussian Processes, again, this misinterpretation took a long time to un-misinterpret

  -  a "hands on" example application in the introduction with example $x$, $x$ and $z$ values would help a clear up much speculation from thee start avoiding many misinterpretations and blind alleys, upon first reading I didn't fully understand the problem setting until the experiments.



I have given a low confidence score to reflect my difficulty in reading the paper.


[3] Variational Learning on Aggregate Outputs with Gaussian Processes, Law et.al. NeurIPS 2019


**Time Spent Reviewing:**

10

---

> ### Author Response · Authors · 2021-08-10
> **Reply to Reviewer szhM**
>
> _“The paper proposes a "conditional mean process", a Gaussian process model that interpolates aggregated data as well as the marginalization operater in order to infer the un-aggregated latent function under this above problem setting.”_
>
> A: “interpolation of aggregated data” may be an inaccurate interpretation of the developed model since equation (5) is rather a linear combination of aggregated data. We would like to emphasize that our contributions also include a non-trivial extension of the original DME paper on a theoretical level by casting it as a reconstruction problem, which provides a different two-staged reconstruction view, thus allowing us to apply convergence results from two-staged regression literature to understand better the learning process of DME.
>
> ------------------------------------------------------------------------------------------------------------------------------------------------------
>
> _“ The integral of a Gaussian process is another Gaussian process is a standard result (eg convolution of a GP is another GP), I feel I have seen variations of equation (3) in multiple previous works [1] [2]..._
>
> _“I believe the following "deconditioning" step is standard (as rightly acknowledged to be Gaussian conditioning) and equations (5), (6) are textbook results. ””_
>
> A: We would like to clarify that we do not claim that the “integral of a GP is another GP” is a novel result. This standard result is just a means for us to compute the cross-covariance between a GP and an integrated GP. The specific form of this cross-covariance and the method to estimate it are one of the contributions in our work. This allows us to recover the result from Hsu and Ramos (2019) in a much simpler way, i.e. without the need to define and perform inference on the graphical model of the task-transformed GP. We will clarify this more and add further appropriate citations in the revised version.
>
> ------------------------------------------------------------------------------------------------------------------------------------------------------
>
> “I believe the main practical novelty mainly lies in using the kernel $\ell_y$ as a means to model away from $\mathbb{P}(X|Y)$ the training data thereby bridging the gap between the unmatched $y$ and $\tilde{y}$, I think this is really cool and the inclusion of VBAGG baseline (where $\ell_y$ is effectively "switched off") demonstrates the efficacy. However this seems rather unsurprising and rather incremental.“
>
> A: We believe that being able to phrase the deconditioning framework from Hsu and Ramos (2019) as a generalisation of VBAGG-like models bridges two disparate research topics in a novel and non-trivial way. Furthermore, as pointed out by the reviewer, the thus obtained generalisation of VBAGG is able to model away from $\mathbb{P}(X|Y)$ using the kernel $\ell_y$. This model improvement is of great significance for practitioners working with earth and climate observations. Meteorological, chemical or spectral covariates are observed at different levels of aggregation. When a mediating field Y is observed – in the climate realm, we can often use climate models data for Y – this “extension of VBAGG” becomes of great importance to learn between covariates having different levels of aggregation.
>
> ------------------------------------------------------------------------------------------------------------------------------------------------------
>
> _“ $z = \mathbb{E}[f(X)|Y] + \epsilon$ does not explicitly state what the expectation is over …”_
>
> A: We thank the reviewer for pointing out this potential confusion. We will clarify and add explicit notation to emphasize what the expectation is over in cases where ambiguity may arise.
>
> ------------------------------------------------------------------------------------------------------------------------------------------------------
>
> _“the word "conditional" is used very frequently and I believe it is referring to the marginal over $X$ using $\mathbb{P}[X|Y]$...”_
>
> A: In a spirit of consistency, we did stick with notations and wordings used in the corresponding literature of Hsu and Ramos (2019). We note that “disaggregation” is another commonly used terminology in measure theory. We will clarify these points to avoid confusion in the revised version.
>
> ------------------------------------------------------------------------------------------------------------------------------------------------------
>
> _”The authors do not appear to explicitly raise possible failure modes, below is one example. If $\ell_y$ is poorly chosen, this may cause the method to perform worse than VBAGG in matched data setting and worse than naive methods”_
>
> A: We thank the reviewer for pointing this out -- as in any kernel method, the choice of the kernel is indeed important (since we utilize GP formalism, $l_y$ parameters can be selected using maximization of marginal likelihood, for example), and this will be clarified in the revised version However, we would like to point out that we did raise possible failure modes of our method. We would like to refer the reviewer to line [400] in the discussion section where we mentioned our deconditional posterior under the paradigm of covariate shift since that might induce uncertainty in estimating the CMP, and thus affect the reconstruction estimates.
>
> ------------------------------------------------------------------------------------------------------------------------------------------------------
>
> _“a "hands on" example application in the introduction with example , and values would help a clear up much speculation from thee start avoiding many misinterpretations and blind alleys, upon first reading I didn't fully understand the problem setting until the experiments.”_
>
> A: We apologize for the difficulties the reviewer has faced reading this work. We appreciate this suggestion and will make sure to clarify and better illustrate the problem presentation.

---

> > ### Comment · Reviewer_szhM · 2021-08-21
> > **Response Response**
> >
> > Thank you for your clarifying comments.
> >
> > Regarding the theoretical contribution and related fields, I am not familiar enough with the related work hence I will leave judgement to other reviewers. After reading their reviews, it seems that they are generally positive hence I will leave my confidence at low and increase my score to a weak accept.
> >
> > Regarding my comment on "incremental practicality", I agree that aggregate data is significant, while I have experience with aggregate data, I struggle to think of times where there was a coupling variable $y$ and when such data was disconnected across two datasets. As mentioned, an illustrative description of a practical use case would vastly help the lay person understand.
> >
> > Thank you for acknowledging my confusion.

---

### Official Review · Reviewer_wwEN · 2021-07-16

**Rating:** 7
**Confidence:** 5

**Summary:**

This paper extends on the work of deconditional mean embeddings (DMEs) and task transformed Gaussian processes (TTGPs) from Hsu and Ramos (2019) from both an application and theoretical stand point.

From a theoretical stand point, the authors developed the framework around deconditional mean embeddings further by three ways: (1) formulating another elegant way to arrive at deconditional posteriors from Hsu and Ramos (2019), (2) establishing deconditional mean operators (DMOs) as vector-valued regressors in a manner that mirrors Grunewalder et al (2012) which highlights its interpretation as a reconstruction operator, and (3) enhancing guarantees on the convergence rate of deconditional mean operators from what was established in Hsu and Ramos (2019) using the setup and results from  Caponnetto and De Vito (2007), Szabó et al (2016), and Singh et al (2019).

From an application stand point, the authors apply DMEs for refining low resolution spatial fields with high resolution information. This is the special case of DMEs and TTGPs where the task dataset consists of collections of low resolution covariates ($\tilde{y}$) and aggregated targets ($\tilde{z}$), and the transformation dataset consists of collections of (another potentially unmatched set of) low resolution covariates ($y$) each matched with bags of high resolution covariates ($^{b}\bf{x}$). The latter constructions regarding the transformation dataset is what makes this a non-trivial special case due to having bagged high resolution covariates, which allows for a slightly different empirical estimator for the cross-covariance operator. This leads to an alternative conditional mean operator (CMO) which they call the conditional mean shrinkage operator. Finally, the authors then apply their work to toy experiments (swiss roll) and downscaling of atmospheric temperature (CMIP6), where to scale it they also derive a variational formulation to approximate the deconditional posterior.


**Limitations And Societal Impact:**

The work is motivated by applications in statistical downscaling with unmatched multi-resolution data. However, its contributions are general in nature in terms of the technique and not limited to this application. In fact, as highlighted in my review, I believe the general theoretical contributions are more significant than the specific application discussed.

The paper does address societal impact in the submission, citing positive benefits in understanding climate phenomenon from unmatched multi-resolution data which can be quite common, as well as potential negative impacts in using this technique to recover sensitive information from low resolution to high resolution where privacy is a concern.

Overall, I agree with the author's self assessment on limitations and societal impact of the paper.

**Main Review:**

Overall, the contributions of the paper is well motivated and well developed. The paper presents non-trivial extensions to the original DME paper on a theoretical level, and at the same time successfully showcases a particularly useful type of application for the theory developed through refining low-resolution spatial fields with high-resolution information.

In particular, of all the theoretical contributions, I find the first contribution on an alternative view to arrive at deconditional posteriors using conditional mean processes very simple and elegant. In Hsu and Ramos 2019, this posterior was derived by defining and performing inference on the graphical model of the task transformed Gaussian process. Instead, this paper arrives at the posterior by first considering the conditional mean process constructed by taking the conditional mean of an original latent Gaussian process which we would like to infer. Due to linearity of the conditional mean, the resulting process is also a Gaussian process, and the authors derive their mean and covariance functions. This naturally leads to the notion of a joint Gaussian field between noisy observations of the resulting Gaussian process after the conditional mean and the original latent Gaussian process, which means the deconditional posterior can be easily derived from Gaussian conditioning. The authors show that this recovers the same deconditional posterior, also termed the posterior of the task transformed Gaussian process, from Hsu and Ramos (2019), establishing another elegant view for interpreting deconditional posteriors. An added bonus is that the authors also explicitly kept a non-zero mean function throughout this derivation.

The second and third contributions are also helpful in the sense that it casts the deconditional problem as a reconstruction problem, which provides a different two-staged reconstruction regression view to the two-staged task transformed regression view originally discussed in Hsu and Ramos (2019). The theorems developed for capturing the convergence rate under this reconstruction loss is well appreciated and indeed non-trivial.

The main weakness of the paper predominantly lies in its presentation, which hopefully should be easy to fix. Understandably, the presentation is dense due to the amount of content and contributions the authors are trying to communicate. However this is especially where the structure and presentation of the paper can vastly improve the communication of the core contributions of the paper. Unfortunately, as it currently stands it can be difficult to appreciate the contributions and their relationships all at once especially because they operate at different levels of generality.

For instance, all three theoretical contributions (as summarised in my summary section) are in fact highly general to the theory of deconditional mean embeddings, and not limited to the core application discussed in the paper, which is refining low-resolution spatial fields with high-resolution information. To me, these contributions are of higher significance compared to the specific application of focus, which is a special case (albeit non-trivial) to the original DME and TTGP setup. For the author's benefit, I would suggest to emphasize the former more than the latter, in contrast to how it is currently communicated. At the very least, it seems strange that the title, abstract, and introduction focuses heavily on setting up this specific application, while the majority of the remainder of the paper develops or extends a theorical framework that is much more general than the application aforementioned.

Furthering this point with a specific example, section 3.3 seems quite out of place in that this is the only sub-section in the major contribution sections (sections 3 and 4) where the multi-resolution application setting comes into play. It seems more appropriate to delay this to Section 5 instead when the specific application is discussed, and keep section 3 and 4 focused on the more general theoretical developments.

[Update] In light of the shared concerns on the paper's clarity and presentation which I agree with as initially noted, I have revised my score to give more weight to this concern, although this does not change my overall stance on the paper.

**Time Spent Reviewing:**

7

---

> ### Author Response · Authors · 2021-08-10
> **Reply to Reviewer wwEN**
>
> _“The main weakness of the paper predominantly lies in its presentation, which hopefully should be easy to fix. Understandably, the presentation is dense due to the amount of content and contributions the authors are trying to communicate. However this is especially where the structure and presentation of the paper can vastly improve the communication of the core contributions of the paper. Unfortunately, as it currently stands it can be difficult to appreciate the contributions and their relationships all at once especially because they operate at different levels of generality.“_
>
>
> A : We appreciate the reviewer’s kind comments on the general framework that was developed in the paper. We acknowledge, as rightfully pointed out, that the intertwining of theoretical and practical contributions operating at different levels of generality undermines the overall presentation. We will restructure the paper to make the relevance of its generality clear and, hopefully, facilitate the appreciation of our contributions for both theoreticians and practitioners. We should point out that our primary motivation was indeed to address this specific application setting with wide-ranging relevance in climate science, but it did require developing this general framework.
>
> We again thank the reviewer for his thorough understanding and appreciation of our work.

---

> > ### Comment · Reviewer_wwEN · 2021-08-26
> > **Presentation may need improvement**
> >
> > Thank you for your response.
> >
> > While I still maintain a positive stance on the significance of the paper's contributions, I also maintain my concern from my initial review that the presentation of the paper needs work. Currently as it stands, it can be quite difficult to follow and reach a decent level of understanding even if one is already familiar with the prior work. Given that this is a shared concern amongst all reviewers, I think it is fair to say that this greatly hinders potential impact and wider adoption of the work. I have revised my score to reflect this. I encourage the authors to make the best of the useful suggestions here to improve the clarity of the paper.

---

> > > ### Author Response · Authors · 2021-08-31
> > > **Working to improve presentation**
> > >
> > > In the light of the rightfully shared concern among reviewers about the paper's presentation, we understand that the current presentation of our work may hinder its impact and adoption, which we take very seriously. As a consequence, we woud like to share with the reviewers that we have taken active steps to incorporate their suggestions, and in particular are working to:
> > >
> > > - Restructure the paper to clarify the separation between theoretical and practical contributions
> > > - Rewrite sections to emphasize the contributions, their relevance and the line of research in which this work fits
> > >
> > > As a matter of fact, we invite the reviewer to read the newly submitted introduction draft to Reviewer KjUw. We hope that this will provide evidence of our commitment to rework the presentation of the article.

---

### Official Review · Reviewer_KjUw · 2021-07-30

**Rating:** 7
**Confidence:** 3

**Summary:**

The authors consider the problem of refining low-res (LR) spatial observations using high-res (HR) covariates.

Basically, the problem is as follows. The analyst observes LR outcome Z, HR covariate X, and LR covariate Y from the process Z=E[f(X)|Y]+e. The analyst wishes to recover the function f.

In more detail: the analyst observes two datasets: D1 and D2. D1 consists of N observations (x_j, y_j) where x_j may be a bag.  D2 consists of M observations (y_j, z_j). The function f is modelled as a function in an RKHS (with Bayesian or frequentist approach).

The authors characterize the Bayesian version of this problem and derive the GP posterior. The authors characterize the frequentist version of this problem and prove minimax optimal finite sample rates.


**Limitations And Societal Impact:**

What assumptions could be violated, and how would that impact conclusions and policy recommendations?

**Main Review:**

Originality: It seems that the estimator previously appears in Hsu and Ramos. Prop 3.2 and Prop 3.3 appear to be new characterizations of the Bayesian approach. Prop 4.1 demonstrates that the estimator is the solution to a 2-stage (frequentist) regression. Theorem 4.2 relates the problem considered in this paper to the nonparametric instrumental variable problem in order to match symbols with an existing theorem.

Quality: The theoretical results are complete and thorough. Why is H_\ell finite dimensional? This seems to be unnecessary when appealing to the quoted theorem. There seems to be a small clash between the theory and empirics. In theory, k is characteristic and ell is finite. In empirics, k and ell are sums of Guassian and Matern kernels, so they are infinite dimensional and not characteristic.

Clarity: It took me a long time to understand the problem set up: what is observed, what is the goal, and what will be estimated. Figure 1 helps, and should be larger. Please improve the introduction to this learning problem. Theorem D.4 should be in the main text since otherwise its assumptions are hidden.

Significance: It is difficult for me to assess the significance of this paper in the spatial observation resolution literature. Within the RKHS literature, the theoretical contributions are modest: the estimator and main theorem are essentially from previous papers, but the Bayesian characterization and connection to nonparametric instrumental variable regression are new and insightful. I think these should be emphasized more. If there is a strong significance for practitioners, I will raise the score.


**Time Spent Reviewing:**

3

---

> ### Author Response · Authors · 2021-08-10
> **Reply to Reviewer KjUw**
>
> *“Why is H_\ell finite dimensional?”*
>
> A: This is to ensure assumption 7 (see line 849 in supplementary material) is satisfied, otherwise the norm of the vector-valued kernel feature map cannot be uniformly bounded in HS norm. The finite dimensionality here is hence not essential and one can investigate other ways to satisfy assumption 7, which will be clarified in the camera-ready version.
>
> ------------------------------------------------------------------------------------------------------------------------------------------------------
>
> *“There seems to be a small clash between the theory and empirics. In theory, k is characteristic and ell is finite. In empirics, k and ell are sums of Guassian and Matern kernels, so they are infinite-dimensional and not characteristic.”*
>
> A: We thank the reviewer for pointing out this misalignment between theory and empirics. We have indeed considered a finite-dimensional RKHS H_\ell in Section 4, in order to apply theory from two-staged regression and quantify the convergence rate of the estimator. Relaxing this assumption would be an interesting direction for future work. Regarding characteristic kernels, since Gaussian and Matérn kernels are both characteristic (this follows e.g. from considering the support of their Fourier transform), the resulting additive kernel is also characteristic.
>
> ------------------------------------------------------------------------------------------------------------------------------------------------------
>
> *“It took me a long time to understand the problem set up: what is observed, what is the goal, and what will be estimated. Figure 1 helps, and should be larger. Please improve the introduction to this learning problem. Theorem D.4 should be in the main text since otherwise its assumptions are hidden.”*
>
> A: We thank the reviewer for their efforts to understand the problem set up and apologize for the lack of clarity. Due to limited space, additional illustrations and details of the theorem’s assumptions had to be relegated to the supplementary material. This will be rectified in the revised version.
>
> ------------------------------------------------------------------------------------------------------------------------------------------------------
>
> *“the estimator and main theorem are essentially from previous papers, but the Bayesian characterization and connection to nonparametric instrumental variable regression are new and insightful.“*
>
> A: The posterior mean of the deconditional posterior indeed recovers the estimator from Hsu and Ramos (2019). We however propose a new and much simpler derivation of this estimator, using the integrated GP formalism, which in addition yields uncertainty estimates. The estimator is furthermore adapted to a multiresolution setting, which is essential for our application domain Similarly, the convergence theorem does indeed use existing principles previously introduced by Caponnetto and De Vito (2007), Szabó et al (2016), and Singh et al (2019). The framework is however different since the objective is to solve a deconditioning problem. As pointed out by Reviewer wwEN, this differs from existing work in the sense that it attempts to solve a reconstruction problem, and the resulting convergence rate is non-trivial.
>
> ------------------------------------------------------------------------------------------------------------------------------------------------------
>
> *“It is difficult for me to assess the significance of this paper in the spatial observation resolution literature. [...] If there is a strong significance for practitioners, I will raise the score.”*
>
> A: Satellite imagery constitutes a major source of data for climate scientists and more generally for practitioners working with earth observations. There are a multitude of satellites in orbit (e.g. MODIS, Sentinel-1, Sentinel-2, Landsat), each with its own spatial resolution, temporal resolution and capturing images over different spectral domains. Hence, different satellites collect snapshots of different places at different times with their own resolutions, and it is unfeasible to match together observations from different devices. In addition, there is often a need to combine these datasets with measurements from the Earth’s surface, which is very difficult to match to satellite data. Using the method introduced in this work could indirectly match coarse observations from a low-resolution satellite such as MODIS with fine observations from a high-resolution satellite such as Landsat. A mediating field would typically consist of climate models simulations, which provide comprehensive spatiotemporal coverage of various meteorological variables (temperature, relative humidity, pressure etc.). By separately matching climate model simulations with low-resolution observations and high-resolution observations, one could apply deconditional downscaling to enhance the resolution of the low-resolution imagery.

---

> > ### Comment · Reviewer_KjUw · 2021-08-11
> > **Finite dimensional RKHS**
> >
> > Thank you for your thorough answers. I do think it is important to preserve the infinite dimensionality of H_\ell; a major motivation of the RKHS approach is its nonparametric (i.e. infinite dimensional) flexibility. Could you clarify other ways to satisfy assumption 7 in this thread?

---

> > ### Comment · Reviewer_KjUw · 2021-08-18
> > **Willing to improve score if two improvements**
> >
> > I would be willing to improve the score to a 7 if the authors can provide two improvements
> >
> > 1. Clarify other ways to satisfy assumption 7 while maintaining H_{\ell} is infinite dimensional
> > 2. Rewrite the introduction (and please paste a new draft below). The improved introduction must explain to a general audience
> > -what is the problem
> > -why is this problem important
> > -what is the solution
> > -what insights are uncovered
> >
> > Here is a starting point:
> >
> > Spatial data are aggregated over predefined regions due to data collection costs and privacy concerns. Nonetheless, empirical research questions may be more specific than aggregated units.
> >
> > The main theorems connect the problem of disaggregation to (1) a Bayesian perspective and (2) nonparametric instrumental variable regression in order to provide strong guarantees

---

> > > ### Author Response · Authors · 2021-08-23
> > > **Finite Dimensional RKHS**
> > >
> > > $$
> > > \newcommand{\cH}{\mathcal{H}}
> > > $$
> > > Relaxing the finite-dimensionality assumption on $\cH_\ell$ for the convergence rate result in Theorem 4.2 is indeed an exciting research direction and we appreciate the reviewer's interest in this matter.
> > >
> > >
> > >
> > > Learning theory for two-staged regression with the 2nd output dimension being infinite-dimensional is an open problem, with recent attempts like in [R1], where they focus on the one-staged regression setting. We believe this is a non-trivial problem which deserves a thorough investigation in a future submission. To support this claim, we propose to examine potential research directions to relax the finite-dimensionality assumption, and highlight their shortcoming and the challenges they raise.
> > >
> > >
> > >
> > > We recall that $\Gamma(f, f') = \langle f, f'\rangle_k Id_{\cH_\ell}$. When $\cH_\ell$ is infinite-dimensional, the identity operator $Id_{\cH_\ell}$ is not trace class anymore. We hence have,
> > > \begin{equation}
> > > ||\Gamma_{\mu_{X|Y=y}}\||^2_{HS(\cH_\ell, \cH_\Gamma)} = Tr(\Gamma(\mu_{X|Y=y}, \mu_{X|Y=y})) = q(y, y)Tr(Id_{\cH_\ell}) = \infty
> > > \end{equation}
> > > which breaks Assumption 7 from Theorem 4.2. As a first approach, it seems natural to examine alternative choices of $\Gamma$ that could allow to verify $||\Gamma_{\mu_{X|Y=y}}||^2_{\operatorname{HS}(\cH_\ell, \cH_\Gamma)} < \infty$.
> > >
> > >
> > >
> > > Let us replace $Id_{\cH_\ell}$ by a positive symmetric linear operator $T : \cH_\ell\to\cH_\ell$, i.e.\
> > > \begin{equation}
> > >     \Gamma : (f, f')\in\cH_k\times\cH_k\mapsto\langle f, f'\rangle_k T\in\mathcal{L}(\cH_\ell).
> > > \end{equation}
> > >
> > > The corresponding canonical features maps are given by $\Gamma_fh = Th\otimes f$, for any $f, h \in\cH_k\times\cH_\ell$. Hence, using the representer theorem, the solution to the ERM problem in (10) from main paper has following form
> > > \begin{equation}
> > >     \hat D = \sum_{j=1}^M\Gamma_{\hat C_{X|Y}\ell_{\tilde y_j}}c_j = \sum_{j=1}^M Tc_j\otimes \hat C_{X|Y}\ell_{\tilde y_j}
> > > \end{equation}
> > > where the vectors $c_1, \ldots, c_M\in\cH_\ell$ can be found as unique solutions of
> > > \begin{align}
> > >     & \sum_{i=1}^M \left(\Gamma(\hat C_{X|Y}\ell_{\tilde y_i}, \hat C_{X|Y}\ell_{\tilde y_j}) + M\epsilon \delta_{ij}\right)c_i = \ell_{\tilde y_j} & 1\leq j\leq M \\
> > >     \Rightarrow & \sum_{i=1}^M \left(\hat q(\tilde y_i, \tilde y_j)T + M\epsilon \delta_{ij}\right)c_i = \ell_{\tilde y_j} & 1\leq j\leq M.
> > > \end{align}
> > >
> > >
> > > With abuse of notation, we can rewrite the latter in matrix form using notations from paper
> > > \begin{equation}
> > >     (\hat{\bf Q} T + M\epsilon Id_{\cH_\ell}){\bf c}^\top = {\bf\Psi_{\tilde y}}^\top \Rightarrow  {\bf c}^\top = (\hat{\bf Q} T + M\epsilon Id_{\cH_\ell})^{-1}{\bf \Psi_{\tilde y}}^\top
> > > \end{equation}
> > > provided invertibility of the operator (granted by regularization).
> > >
> > >
> > > Plugging things back into the expression of $\hat D$, we obtain
> > > \begin{equation}
> > >     \hat D  = [T{\bf c}]\otimes [\hat C_{X|Y}{\bf \Psi_{\tilde y}}]
> > > \end{equation}
> > > \begin{equation}
> > >  = \left[T{\bf \Psi_{\tilde y}}({\bf \hat Q} T + M\epsilon Id_{\cH_\ell})^{-1}\right]\otimes \left[\hat C_{X|Y}{\bf\Psi_{\tilde y}}\right]
> > > \end{equation}
> > > \begin{equation}
> > >  =T{\bf\Psi_{\tilde y}}(\hat{\bf Q} T + M\epsilon Id_{\cH_\ell})^{-1}{\bf\Psi_{\tilde y}}^\top \hat C_{X|Y}^\top
> > > \end{equation}
> > > \begin{equation}
> > >  =T{\bf\Psi_{\tilde y}}(\hat{\bf Q} T + M\epsilon Id_{\cH_\ell})^{-1} {\bf A\Phi_x}.
> > > \end{equation}
> > >
> > >
> > >
> > > If we choose $T = Id_{\cH_\ell}$, we verify that we recover the DMO empirical estimator $\hat D = \hat D_{X|Y}$. On the other hand, if $T$ is chosen to be a trace class operator, we then verify
> > > \begin{equation}
> > >     ||\Gamma_{\mu_{X|Y=y}}||^2_{HS(\cH_\ell, \cH_\Gamma)} = q(y, y) Tr(|T|) < \infty.
> > > \end{equation}
> > >
> > >
> > > Hence Assumption 7 is satisfied and Theorem 4.2 holds for an infinite-dimensional $\cH_\ell$. For example, one may construct such trace class operator $T$ out of the cross-covariance operators $C_{XY}$ and $C_{YY}$ – which are Hilbert-Schmidt operators. However, we note that the solution to the vector-valued regression problem may not necessarily correspond to the empirical DMO estimator of Hsu & Ramos anymore. While this approach would lead to other estimators for the deconditioning problem, which would be an interesting topic for future research, the connection to previous work by Hsu & Ramos, as well as to our own Gaussian process formalism is no longer immediate and requires further study via the appropriate choice of $T$.
> > >
> > >
> > >
> > >
> > > Alternatively, a choice of $T$ that would allow to recover the right estimator is
> > > \begin{equation}
> > >     \Gamma(f, f') = \langle f, f'\rangle_k proj_{Span\\{\ell_{\tilde y_j}\\}}
> > > \end{equation}
> > > where $proj_{Span\\{\ell_{\tilde y_j}\\}}$ denotes the projection over the vector space spanned by $\{\ell_{\tilde y_1}, \ldots, \ell_{\tilde y_M}\}$. Since this choice of kernel does not interfere with the optimization problem, it yields the exact same solution $\hat D = \hat D_{X|Y}$. However, we have
> > >
> > >
> > > \begin{equation}
> > >     ||\Gamma_{\mu_{X|Y=y}}||^2_{HS(\cH_\ell, \cH_\Gamma)} = q(y, y)Tr\left({proj_{Span\\{\ell_{y_j}\\}}}\right) \leq q(y, y) dim\left(Span\\{\ell_{y_j}\\}\right) \leq q(y,y) M.
> > > \end{equation}
> > >
> > >
> > > Hence, the bound is not uniform which breaks Assumption 7. In particular, the assumptions from [Theorem 5, R2] are not satisfied anymore. An interesting direction however would be to incorporate the $\mathcal{O}(M)$ bound in the derivation of [Theorem 5, R2] to study whether one can obtain – potentially slower – convergence guarantees.
> > >
> > >
> > >
> > > We hope this clarifies to what extent the finite-dimensionality of $\cH_\ell$ is only required to obtain the uniform boundedness of the family of operators $\\{\Gamma_{\mu_{X|Y=y}}\mid y\in\mathcal{Y}\\}$. Making a weaker assumption that allows infinite dimensional $\cH_\ell$ while bounding $\\{\Gamma_{\mu_{X|Y=y}}\mid y\in\mathcal{Y}\\}$ is thus possible. As shown above, it however stands as a challenging problem which we believe could be the focus of a future submission and is out of the scope of the current project.
> > >
> > > We thank the reviewer for their patience and will get back shortly with a draft of the rewritten introduction.
> > >
> > >
> > > [R1]  Park, Junhyunng, and Krikamol Muandet. "Regularised Least-Squares Regression with Infinite-Dimensional Output Space." arXiv preprint arXiv:2010.10973 (2020).
> > >
> > > [R2] Z. Szabó, B. K. Sriperumbudur, B. Póczos, and A. Gretton, “Learning theory for distribution regression,” J. Mach. Learn. Res., vol. 17, pp. 1–40, 2016.

---

> > > > ### Comment · Reviewer_KjUw · 2021-08-31
> > > > **diving deeper**
> > > >
> > > > I think there is a difference between how Gamma is defined in this work versus Szabo et al, which leads to this difference in assumptions. Here, Gamma_{mu X|Y=y}: H_{\ell} --> H_{Gamma}. In Szabo et al, Gamma_{mu X| Y=y}: \mathcal{Z} --> H_{Gamma}.
> > > >
> > > > The desired condition is eq (11) is Szabo et al for their definition of Gamma. In eq (47) of Szabo et al, the authors state that it is sufficient for the outcome variable to be a real scalar. In this paper's notation, that is saying Z is a real scalar, which the authors state on line 95 of this paper. However, Gamma is defined differently in this work than in Szabo et al.
> > > >
> > > > I appreciate the discussion provided above. I found Appendix A.4.4 of Singh et al to be an interesting discussion on related topics as well
> > > >
> > > > Overall, there is a subtle difference in the RKHS constructions. Mentioning this difference, and that it could possibly be bridged in future work, would be great

---

> > > ### Author Response · Authors · 2021-08-31
> > > **Introduction Draft**
> > >
> > > **We thank the reviewer for their patience. Please find a first draft of the introduction below. We hope this new draft will clarify the problem relevance in climate science, how it differs from usual statistical downscaling setups, the proposed solution and contributions. A color code has been used to indicate the beginning (¶) and ending (.) of sentences answering each of the following questions:**
> > > - $\colorbox{yellow}{What is the problem?}$
> > > - $\colorbox{orange}{Why is this problem important?}$
> > > - $\colorbox{turquoise}{What is the solution?}$
> > > - $\colorbox{pink}{What insights are uncovered?}$
> > >
> > > ---
> > >
> > > $\colorbox{yellow}{¶}$Spatial observations often operate at limited resolution due to practical constraints. For example,
> > > remote sensing atmosphere products [1,2,3,4] provide measurements of atmospheric properties such as cloud top temperature and optical thickness, but at low resolution only due to instrumental limitations$\colorbox{yellow}{.}$$\colorbox{orange}{¶}$ Devising methods to refine low-resolution (LR) variables for local-scale analysis thus plays a crucial part in our understanding of the anthropogenic impact on climate$\colorbox{orange}{.}$
> > >
> > > $\colorbox{turquoise}{¶}$When high-resolution (HR) observations of different covariates are available, details can be instilled into the LR field for refinement. This task is referred to as *statistical downscaling* and models LR observations as the aggregation of an unobserved underlying HR field$\colorbox{turquoise}{.}$ $\colorbox{orange}{¶}$For example, multispectral optical satellite imagery [5,6] typically comes at higher resolution than atmospheric products and can be used for refining the latter$\colorbox{orange}{.}$
> > >
> > > $\colorbox{turquoise}{¶}$Statistical downscaling has been studied in various forms, notably giving it a probabilistic treatment [7,8,9,10,11,12], in which Gaussian processes (GP) [13] are typically used in conjunction with a sparse variational formulation [14] to recover the underlying unobserved HR field. Our approach follows this line of research where we do not observe data from the underlying HR groundtruth field. On the other hand, deep neural network (DNN) based approaches [15,16,17] study this problem from a different setting, where they often assume that both HR and LR matched observations are available for training. Then, their approaches follow a standard supervised learning setting in learning a mapping between different resolutions$\colorbox{turquoise}{.}$
> > >
> > > $\colorbox{yellow}{¶}$However, both lines of existing methods require access to bags of HR covariates that are paired with aggregated targets, which in practice might be infeasible$\colorbox{yellow}{.}$$\colorbox{orange}{¶}$ For example, the multitude of satellites in orbit not only collect snapshots of the atmosphere at different resolutions, but also from different places and at different times, such that these observations are not jointly observed$\colorbox{orange}{.}$$\colorbox{turquoise}{¶}$ To overcome this limitation, we propose to consider a more flexible *Mediated Statistical Downscaling* setup that only requires indirect matching between LR and HR covariates through a mediating field. We assume that this additional field can be easily observed, and matched separately with both HR covariates and aggregate LR targets. We then use this third-party field to mediate learning and downscale our unmatched data. In our motivating application, climate simulations [18,19,20] based on physical science can serve as a mediating field since they provide a comprehensive coverage of meteorological variables that can be matched to both LR and HR covariates$\colorbox{turquoise}{.}$
> > >
> > > $\colorbox{turquoise}{¶}$Formally, let $^b\\!\boldsymbol{x} = \\{x^{(1)}, \ldots, x^{(n)}\\}\subset\mathcal{X}$ be a general notation for bags of HR covariates, $f:\mathcal{X}\to\mathbb{R}$ the field of interest we wish to recover and $\tilde z$ the LR aggregate observations from the field $f$. We suppose that $^b\\!\boldsymbol{x}$ and $\tilde z$ are unmatched, but that there exists mediating covariates $y, \tilde y\in\mathcal{Y}$, such that $(^b\\!\boldsymbol{x}, y)$ are jointly observed and likewise for $(\tilde y, \tilde z)$ as illustrated in Figure 1. We assume the following aggregation observation model $\tilde z = \mathbb{E}[f(X)|Y=\tilde y] + \varepsilon$ with some noise $\varepsilon$. Our goal in mediated statistical downscaling is then to estimate $f$ given $(^b\\!\boldsymbol{x}, y)$ and $(\tilde y, \tilde z)$, which corresponds to the *deconditioning* problem introduced in [21]$\colorbox{turquoise}{.}$
> > >
> > > Motivated by applications in likelihood-free inference and task-transfer regression, Hsu and Ramos [21] first studied the deconditioning prob-lem through the lens of reproducing kernel Hilbert space (RKHS) and introduced the framework of Deconditional Mean Embeddings (DME) as its solution. $\colorbox{pink}{¶}$In this work, motivated by our mediated statistical downscaling problem, we extend deconditioning to a multi-resolution setup. By placing a GP prior on the sought field $f$, we obtain a posterior distribution of the downscaled field as a principled Bayesian solution to the downscaling task on indirectly matched data. Our formulation results into a much simpler and elegant way to arrive to the DME-based estimator of Hsu and Ramos [21], using the conditional expectations of $f$. For scalability, we provide a tractable variational inference approximation and an alternative approximation to the conditional mean operator (CMO) [22] to speed up computation for large multi-resolution datasets$\colorbox{pink}{.}$
> > >
> > > $\colorbox{pink}{¶}$From a theoretical stand point, we further develop the framework of deconditional mean embeddings by establishing it as a two-staged vector-valued regressor with a natural reconstruction loss that mirrors Grünewälder et al. [23]’s work on conditional mean embeddings. This perspective allows us to leverage distribution regression theory from [24,25] and obtain novel convergence rate results for the deconditional mean operator (DMO) estimator. Under mild assumptions, we obtain conditions under which this rate is a minimax optimal in terms of statistical-computational efficiency$\colorbox{pink}{.}$
> > >
> > > Our contributions are summarized as follows:
> > > - We propose a Bayesian formulation of the mediated statistical downscaling problem and establish a connection to the deconditioning problem of Hsu and Ramos [21]. We establish its posterior mean as a DME-based estimate and its posterior covariance as a gauge of the deconditioning quality. Computationally efficient algorithms are devised.
> > > - We demonstrate that the DMO estimate minimises a two-staged vector-valued regression and derive its convergence rate under mild assumptions, with conditions for minimax optimality.
> > > - We benchmark our model against existing methods for spatial disaggregation tasks in climate science, on both synthetic and real-world multi-resolution atmospheric fields data, and show improved performance.

---

> > > > ### Comment · Reviewer_KjUw · 2021-08-31
> > > > **improvement**
> > > >
> > > > Thank you for the improved introduction. For both improvements (introduction and discussion of finite dimensional RKHS) I am raising the score

---

### Author Response · Authors · 2021-08-10
**General response to the reviewers**


We would like to thank the reviewers for their careful reading and the expertise they invested in this review. We have taken their thoughtful comments on board to improve and clarify our work. Here are two messages we would like to highlight:

### 1. Contributions
(**Theoretical contribution for the Deconditioning mean operator**) As reviewer wwEN kindly elaborated, we formulate an elegant way to recover the deconditional RKHS-based estimator from Hsu & Ramos [2019] using a Gaussian Process formalism and cast the deconditional mean operators (DMOs) as vector-valued regressors which allow us to quantify the convergence rate using established theories from the two-staged learning theory literature under mild assumptions. These are indeed non-trivial contributions to the theory of deconditional mean embeddings.

(**Practical contribution to climate science**) On the practical side, we would like to highlight the significance of this problem set up for the climate science community. When working with spatiotemporal observations – as is typically the case in climate science – it is common to collect observations from different devices. Each device has its own spatial and temporal resolution, such that they capture information about different locations, at different times and different resolutions. It is thus impossible to model these samples as jointly observed. Climate models on the other hand provide comprehensive simulation data of meteorological variables (temperature, pressure etc.) at any time and place on Earth. This simulation data can be separately matched with observations from each measurement device and serve as a mediating field. The method proposed in this paper can then be used to downscale observations with the lowest resolution onto observations with the highest resolution. This is of great significance for practitioners since it enables learning between unmatched multiresolution observations – which are expensive to acquire – using simulated data as mediation – which is much cheaper to obtain.

### 2. Presentations
We are thankful for the reviewers’ efforts to understand the problem setup and the paper’s contributions. We apologize for the parts where clarity needed improving and that some reviewers faced difficulties reading this work due to the need to communicate in the paper both the theoretical contributions as well as the contributions with a more application-driven focus. We will ensure to incorporate the reviewers’ suggestions in the camera-ready version and improve the paper’s readability. Namely, we will better clarify and illustrate the problem presentation, correct sources of ambiguity, and restructure the paper to clarify the distinction between contributions to the theory of deconditional mean embeddings and contributions to statistical downscaling.

We again thank the reviewers for their time to review our work and hope that the above and individual responses will clarify their concerns and hope the reviewers can revise their scores accordingly.

---

### Decision · Program_Chairs · 2021-09-27

**Decision:**

Accept (Poster)

**Comment:**


After the rebuttal and discussion, the consensus among the high-confidence reviewers is that the paper should be accepted. The meta-reviewer agrees. On the theoretical side, the paper makes valuable contributions that clearly go beyond previous work. On the practical side, the application in climate science is valuable.

For the community to fully benefit from the paper, serious efforts need to go into making the paper more accessible. First, the explanation of the problem setup and the background needs to be improved to make the paper accessible to non-experts. In addition to the advice given by the reviewers, which needs to be included, the authors could make use of the appendix to further clarify the background material and previous work. Second, as promised by the authors in the rebuttal, the presentation of the two theoretical contributions to (1) the theory of deconditional mean embeddings and (2) statistical downscaling needs to be improved.